# Near-Optimal Sample Complexities of Divergence-based S-rectangular Distributionally Robust Reinforcement Learning

## Abstract

Distributionally robust reinforcement learning (DR-RL) has recently gained significant attention as a principled approach that addresses discrepancies between training and testing environments. To balance robustness, conservatism, and computational traceability, the literature has introduced DR-RL models with SA-rectangular and S-rectangular adversaries. While most existing statistical analyses focus on SA-rectangular models, owing to their algorithmic simplicity and the optimality of deterministic policies, S-rectangular models more accurately capture distributional discrepancies in many real-world applications and often yield more effective robust randomized policies. In this paper, we study the empirical value iteration algorithm for divergence-based S-rectangular DR-RL and establish near-optimal sample complexity bounds of $\widetilde{O}(|\mathcal{S}||\mathcal{A}|(1-\gamma)^{-4}\varepsilon^{-2})$, where $\varepsilon$ is the target accuracy, $|\mathcal{S}|$ and $|\mathcal{A}|$ denote the cardinalities of the state and action spaces, and $\gamma$ is the discount factor. To the best of our knowledge, these are the first sample complexity results for divergence-based S-rectangular models that achieve optimal dependence on $|\mathcal{S}|$, $|\mathcal{A}|$, and $\varepsilon$ simultaneously. We further validate this theoretical dependence through numerical experiments on a robust inventory control problem and a theoretical worst-case example, demonstrating the fast learning performance of our proposed algorithm.

## 1   Introduction

Reinforcement learning (RL) Sutton and Barto [20] is a powerful machine learning framework in which agents learn to make optimal sequential decisions through continuous interaction with an environment. While RL has achieved remarkable success across various domains, its practical deployment faces a significant challenge: real-world deployment conditions often differ from the training environment (e.g., simulations), resulting in fragile policies that fail to generalize. This mismatch undermines RL's applicability in real-world settings, where discrepancies between training and deployment are the norm.

The framework of distributionally robust reinforcement learning (DR-RL) was thus proposed in Zhou et al. [32] to address this mismatch and has since been further developed in a series of works, including Panaganti and Kalathil [13], Yang et al. [30], Xu et al. [28], Blanchet et al. [1], Liu et al. [10], Wang et al. [21], Yang et al. [31], Wang et al. [25], Shi and Chi [17].

Popular models in distributionally robust reinforcement learning (DR-RL) include those based on SA-rectangular and S-rectangular uncertainty sets. The notion of *rectangularity*, originally introduced in the robust MDP literature to describe the adversary's temporal flexibility in selecting distributions [8], has since been refined. With the incorporation of various information structures and a growing

focus on constraining adversarial power, rectangularity now serves to impose structural limitations on uncertainty sets, as elaborated in Le Tallec [9] and Wiesemann et al. [26]. In particular, SA-rectangularity allows the adversary to choose separate distributions for each state-action pair, whereas S-rectangularity enforces consistency across actions within a given state, thereby offering a more confined modeling choice.

Existing statistical analyses of DR-RL predominantly focus on the SA-rectangular setting, primarily due to its computational tractability. Moreover, it has been shown that SA-rectangular models always admit deterministic optimal policies. However, as illustrated in the example below, the S-rectangular formulation can be more appropriate and less conservative in certain applications, such as inventory management.

**Example 1** (Inventory Model). Consider a classical inventory control problem where the inventory evolves according to $S_{t+1} = S_t + A_t - D_t$, with $\{D_t : t \geq 0\}$ representing the stochastic demand process and $a_t$ denoting the replenishment decision at time $t$. The reward function is $R(S_t, A_t, S_{t+1}) = p(S_t - S_{t+1} + A_t) + b \min(S_{t+1}, 0) - h \max(S_{t+1}, 0) - cA_t$, where $p$ is the sales price, $c$ is the purchase cost, $h$ is the holding cost, and $b$ is the penalty of backlog. To address the uncertainty in demand, distributionally robust reinforcement learning (DR-RL) provides a natural framework for enhancing robustness. In this context, it is reasonable to assume that the adversary can only modify the distribution of the demand $D_t$ independently of the controller's action $A_t$, leading to an S-rectangular uncertainty set. By contrast, the SA-rectangular formulation allows the adversary to choose different distributions for $D_t$ based on the controller's action $A_t$—for example, assigning low demand when $A_t$ is large and high demand when $A_t$ is small—granting the adversary excessive power and resulting in an unrealistic model.

This example highlights how S-rectangularity constrains the adversary's power by preventing it from adapting to the controller's actions, making it a more practical and less conservative modeling choice in applications such as inventory management.

While suitable for many applications, the S-rectangular formulation in DR-RL is more challenging than its SA-rectangular counterpart, both statistically and computationally, due to the possibility of randomized optimal policies. Computationally, this requires solving a full min-max problem rather than a simpler maximization. Fortunately, Ho et al. [7] proposed an efficient method for performing Bellman updates in this setting. Statistically, the challenge arises from the fact that the space of randomized policies is exponentially larger than the space of deterministic policies typically sufficient under SA-rectangularity.

Another feature of Example 1 is that the reward depends on the current state $S_t$, the current action $A_t$, and the next state $S_{t+1}$. In contrast, the literature typically considers reward functions of the form $R(S_t, A_t)$, which depend only on the current state and action. The inventory management example highlights the necessity of adopting a reward function of the form $R(S_t, A_t, S_{t+1})$ to accurately capture the underlying dynamics.

In this work, we study the problem of learning the optimal value function in a divergence-based S-rectangular robust MDP, where the uncertainty set is defined as the sum of divergences across all actions. This formulation is well motivated in practice, as divergence-based uncertainty sets preserve absolute continuity and are widely adopted in the literature [7, 30], where efficient algorithms for computing the robust value function have been developed.

However, a satisfactory analysis of the minimax statistical complexity for learning the value function remains missing. To the best of our knowledge, the current state-of-the-art upper bound in Yang et al. [30] contains a sample complexity dependence on $|\mathcal{S}|$ and $|\mathcal{A}|$ in the form of $O(|\mathcal{S}|^2|\mathcal{A}|^2)$, where $|\mathcal{S}|$ and $|\mathcal{A}|$ are the cardinalities of the state and action spaces. This significantly deviates from the known lower bound of $\Omega(|\mathcal{S}||\mathcal{A}|)$. In addition, we have pointed out that in many models of practical interest (e.g., Example 1), the reward function depends naturally on the next state $S_{t+1}$, a structural feature that is often overlooked in the existing sample complexity literature.

We contribute to the literature by analyzing divergence-based S-rectangular robust MDPs with reward functions that depend on the current state, current action, and next state, i.e., $R(S_t, A_t, S_{t+1})$. We establish a sample complexity bound of $\widetilde{O}(|\mathcal{S}||\mathcal{A}|(1-\gamma)^{-4}\varepsilon^{-2})$, where $\varepsilon$ is the target accuracy and $\gamma$ is the discount factor. This bound is optimal in its dependence on $|\mathcal{S}|$, $|\mathcal{A}|$, and $\varepsilon$, and it holds uniformly over the entire range of uncertainty sizes $\rho \in (0, +\infty)$ and discount factors $\gamma \in (0, 1)$.

89  To the best of our knowledge, this is the first sample complexity upper bound for divergence-based
90  S-rectangular models that simultaneously achieves optimal dependence on $|\mathcal{S}|$, $|\mathcal{A}|$, and $\varepsilon$.

91  To achieve the optimal $|\mathcal{S}||\mathcal{A}|$ dependence, we develop a refined sensitivity analysis that improves
92  upon the metric entropy bounds derived from the covering numbers of the randomized policy
93  class $\Pi = \{(\pi(\cdot|s))_{s \in \mathcal{S}} \mid \pi(\cdot|s) \in \Delta(\mathcal{A})\}$, where $\Delta(\mathcal{A})$ denotes the probability simplex over $\mathcal{A}$,
94  as used in Yang et al. [30]. Moreover, our analyses advance the techniques of Wang et al. [25] by
95  relaxing the mutual absolute continuity requirement, thereby extending the allowable range of the
96  uncertainty radius to $\mathbb{R}+$, beyond the previously restrictive regime of $\rho = O(\mathfrak{p}_\wedge)$, while retain an
97  $O(1)$ dependence on $\rho$ as $\rho \downarrow 0$.

98  The remainder of this paper is organized as follows: Secction 2 briefly reviews related work on SA-
99  rectangular and S-rectangular distributionally robust reinforcement learning. Section 3 introduces the
100  framework for learning S-rectangular distributionally robust Markov Decision Processes. Section 4
101  establishes sample complexity upper bounds for value function estimation. Section 5 presents
102  numerical experiments to support our theoretical results.

## 2   Literature Review

104  In this section, we briefly survey SA-rectangular and S-rectangular distributionally robust reinforce-
105  ment learning.

106  **SA-rectangular DR-RL:** The dynamic programming principles for SA-rectangular distributionally
107  robust Markov decision processes (DR-MDPs) have been gradually established through a series of
108  works under different information structures [6, 8, 12, 15, 24]. Recent advances in SA-rectangular
109  distributionally robust reinforcement learning (DR-RL) have explored sample complexity in various
110  settings. Broadly speaking, model-based approaches have been studied in Zhou et al. [33], Panaganti
111  and Kalathil [13], Yang et al. [30], Shi and Chi [16], Xu et al. [28], Shi et al. [18], Blanchet et al. [1],
112  while the statistical properties of model-free algorithms are presented in Liu et al. [10], Wang et al.
113  [21, 22], Yang et al. [31].

114  **S-rectangular DR-RL:** To extend the flexibility of robust MDP models, S-rectangularity was
115  introduced in Xu and Mannor [27], Wiesemann et al. [26] as an overarching theoretical framework to
116  constrain the adversary while retaining a dynamic programming equation. Ho et al. [7] developed an
117  efficient optimization algorithm to solve the Bellman update under this structure. On the statistical
118  side, Yang et al. [30] provided the first sample complexity result for S-rectangular DR-RL, achieving
119  a rate of $\widetilde{O}(|\mathcal{S}|^2|\mathcal{A}|^2(1-\gamma)^{-4}\varepsilon^{-2})$, which is suboptimal in its dependence on the number of states
120  and actions. More recently, Clavier et al. [2] established near-optimal rates for the S-rectangular
121  setting under general $L_p$ norm uncertainty sets. However, their analysis does not directly extend to
122  divergence-based uncertainty sets.

## 3   Learning S-rectangular Robust Markov Decision Processes

### 3.1   Classical Markov Decision Processes

125  We briefly review and establish notation for classical tabular MDP models. Let $\Delta(\mathcal{S}), \Delta(\mathcal{A})$ denote
126  the probability simplex over the finite state space $\mathcal{S}$ and action space $\mathcal{A}$ respectively. An infinite hori-
127  zon MDP is defined by the tuple $(\mathcal{S}, \mathcal{A}, R, P, \gamma)$, where $\mathcal{S}$ and $\mathcal{A}$ are the finite state and action spaces,
128  respectively; $R : \mathcal{S} \times \mathcal{A} \times \mathcal{S} \to [0, 1]$ is the reward function; $P = \{P_{s,a}(\cdot) \in \Delta(\mathcal{S}) : (s, a) \in \mathcal{S} \times \mathcal{A}\}$
129  is the controlled transition kernel; and $\gamma \in (0, 1)$ is the discount factor. Through out the paper,
130  given a controlled transition kernel $P$, we denote $P_s := (P_{s,a})_{a \in \mathcal{A}}$ which is seen as a function
131  $P_s : A \to \Delta(\mathcal{S})$.

132  We define the measurable space $(\Omega, \mathcal{F})$ to be the canonical space $(\mathcal{S} \times \mathcal{A})^{\mathbb{N}}$ equipped with the
133  $\sigma$-field generated by cylinder sets. Define state-action process $(S_t, A_t)_{t \geq 0}$ by the point evaluation
134  $X_t(\omega) = s_t, A_t(\omega) = a_t$ for all $t \geq 0$ and $\omega = (s_0, a_0, s_1, a_1, \dots) \in \Omega$.

135  An agent may optimize over the class of history-dependent policies, denoted by $\Pi_{\mathrm{HD}}$, where each
136  policy $\pi = (\pi_t)_{t \geq 0} \in \Pi_{\mathrm{HD}}$ is a sequence of decision rules. Each decision rule $\pi_t$ at time $t$
137  specifies the conditional distribution of the action $A_t$ given the full history, that is, a mapping
138  $\pi_t : (\mathcal{S} \times \mathcal{A})^t \times \mathcal{S} \to \Delta(\mathcal{A})$. In the setting of classical infinite-horizon discounted MDPs, it is well

known that optimal decision-making can be achieved using stationary, Markov, deterministic policies, denoted $\Pi_D$, where each policy is a mapping $\pi : \mathcal{S} \to \mathcal{A}$ [14].

However, in the context of S-rectangular DRMDPs, policies in $\Pi_D$ may fail to attain the optimal performance achievable within the broader class $\Pi_{HD}$ [26]. In this setting, it suffices to consider stationary, Markov, randomized policies, which we denote by $\Pi$ throughout the paper. Each $\pi \in \Pi$ is a mapping $\pi : \mathcal{S} \to \Delta(\mathcal{A})$, specifying a conditional distribution over actions given the current state $S_t$, uniformly for all $t \geq 0$. Given this sufficiency, we restrict our attention to policies in the class $\Pi$ for the remainder of the paper.

Given a controlled transition kernel $P$ of a classical MDP, a policy $\pi \in \Pi$ and an initial distribution $\mu \in \Delta(\mathcal{S})$ uniquely defines a probability measure on $(\Omega, \mathcal{F})$. We will always assume that $\mu$ is the uniform distribution over $\mathcal{S}$. The expectation under this measure is denoted by $E_P^\pi$. The infinite horizon discounted value $V_P^\pi$ is defined as:

$$V_P^\pi(s) := E_P^\pi \left[ \sum_{t=0}^\infty \gamma^t R(S_t, A_t, S_{t+1}) \middle| S_0 = s \right].$$

An optimal policy $\pi^* \in \Pi$ achieves the optimal value $V_P^*(s) := \max_{\pi \in \Pi} V_P^\pi(s)$.

It is well known that the optimal value function is the unique solution of the following *Bellman equation*:

$$v(s) = \max_{a \in A} \sum_{s' \in \mathcal{S}} P_{s,a}(s')(R(s,a,s') + \gamma v(s')). \tag{3.1}$$

Let $v^*$ be the unique solution, then any deterministic policy $\pi^* : \mathcal{S} \to \mathcal{A}$ with $\pi^*(s) \in \arg\max_{a \in \mathcal{A}} \sum_{s' \in \mathcal{S}} P_{s,a}(s')(R(s,a,s') + \gamma v^*(s'))$ will achieve the optimal value $V_P^*(s)$.

## 3.2 Robust MDPs and S-Rectangularity

Robust MDPs extend standard MDP models by introducing an adversary that perturbs the transition dynamics within a prescribed uncertainty set $\mathcal{P}$, aiming to minimize the control value achieved by the decision maker. This formulation gives rise to a dynamic zero-sum game between the controller and the adversary. Consequently, the controller must account for potential model misspecifications represented by the adversary perturbation, leading to the design of more robust policies.

The statistical complexity of policy learning in robust MDPs has been primarily studied under SA- and S-rectangular uncertainty sets. As discussed in the previous section, S-rectangularity generalizes SA-rectangular models and provides a more expressive framework for modeling adversarial perturbations, constraining the adversary in a structured way while preserving the dynamic programming principle. From this point forward, we will be focusing on S-rectangular robust MDPs.

**Definition 1** (Wiesemann et al. [26], S-rectangularity)**.** The uncertainty set $\mathcal{P}$ is S-rectangular if $\mathcal{P} = \bigtimes_{s \in \mathcal{S}} \mathcal{P}_s$ for some $\mathcal{P}_s \subseteq \{(\psi_a)_{a \in \mathcal{A}} | \psi_a \in \Delta(\mathcal{S}), \forall a \in \mathcal{A}\}$ for all $s \in \mathcal{S}$.

We focus on a special class of S-rectangular adversarial uncertainty sets, where the controlled transition kernels are perturbations of a nominal kernel $\overline{P}$. These sets are defined via a divergence function $f$ and a radius parameter $\rho$. The computational methods and statistical complexity associated with this type of uncertainty structure have been extensively studied in the literature [30, 7].

Specifically, given a divergence function $f$, i.e. $f : \mathbb{R}_+ \to \mathbb{R}$ is convex with $f(0) = 1$ and $f(0) = \lim_{t \downarrow 0} f(t)$, we consider the S-rectangular uncertainty set $\mathcal{P}(f, \rho) = \bigtimes_{s \in \mathcal{S}} \mathcal{P}_s(f, \rho)$ under $f$-divergence and radius $\rho$ where

$$\mathcal{P}_s(f, \rho) = \left\{ P_{s,a} \in \Delta(\mathcal{S}) \middle| P_{s,a} \ll \overline{P}_{s,a}, \sum_{s' \in \mathcal{S}, a \in \mathcal{A}} \overline{P}_{s,a}(s') f\left(\frac{P_{s,a}(s')}{\overline{P}_{s,a}(s')}\right) \leq |\mathcal{A}|\rho \right\}. \tag{3.2}$$

Here, $\ll$ denotes absolute continuity; i.e. a probability measure $p \in \Delta(\mathcal{S})$ is absolutely continuous with respect to $q \in \Delta(\mathcal{S})$, denoted by $p \ll q$, if $q(s) = 0$ implies $p(s) = 0$ for any $s \in \mathcal{S}$. The dependence of the uncertainty set on $(f, \rho)$ is suppressed when there is no ambiguity.

Given a policy $\pi \in \Pi_{HD}$ and uncertainty set $\mathcal{P} = \mathcal{P}(f, \rho)$, the robust value function of $\pi$ is

$$V_{\mathcal{P}}^\pi(s) = \inf_{P \in \mathcal{P}} E_P^\pi \left[ \sum_{t=0}^\infty \gamma^t R(S_t, A_t, S_{t+1}) \middle| S_0 = s \right] \tag{3.3}$$

for all $s \in \mathcal{S}$. The optimal value, defined as $V^*_\mathcal{P}(s) := \sup_{\pi \in \Pi_{\text{HD}}} V^\pi_\mathcal{P}(s)$, is achieved by $\pi^* \in \Pi$.

**Definition 2** (DR Bellman Equation). Given S-rectangular $\mathcal{P} = \times_{s \in \mathcal{S}} \mathcal{P}_s$, the DR Bellman equation is the following fixed-point equation for $v : \mathcal{S} \to \mathbb{R}$

$$v(s) = \sup_{\phi \in \Delta(\mathcal{A})} \inf_{P_s \in \mathcal{P}_s} \sum_{a \in \mathcal{A}} \phi(a) \left[ \sum_{s' \in \mathcal{S}} P_{s,a}(s') \left( R(s,a,s') + \gamma v(s') \right) \right]. \tag{3.4}$$

It is well known [26] that for $\mathcal{P} = \mathcal{P}(f, \rho)$ the optimal value $V^*_\mathcal{P}$ is the unique solution $v^*$ to (3.4).

We note that value function in (3.3) assumes an adversary that fixes a controlled transition kernel over the entire control horizon, a setting commonly referred to as a static or time-homogeneous adversarial model [8, 26, 24]. This framework can be extended to more general Markovian or history-dependent adversarial models, while still preserving Markov optimality [24].

To facilitate our analysis, we define the DR Bellman operators as follows.

**Definition 3** (DR Bellman Operators). Given uncertainty set $\mathcal{P} = \mathcal{P}(f, \rho)$ and $\pi \in \Pi$ the (population) DR Bellman operator is defined as

$$\mathcal{T}^\pi(v)(s) := \inf_{P \in \mathcal{P}} \left( \sum_{a \in \mathcal{A}} \pi(a|s) \left[ \sum_{s' \in \mathcal{S}} P_{s,a}(s') \left( R(s,a,s') + \gamma v(s') \right) \right] \right) \tag{3.5}$$

for all $s \in \mathcal{S}$. The optimal DR Bellman operator is $\mathcal{T}^*(v)(s) := \sup_{\pi \in \Pi} \mathcal{T}^\pi(v)(s), \forall s \in \mathcal{S}$.

### 3.3 Generative Model and the Empirical Bellman Estimator

The sample complexity analysis in this paper assumes the availability of a *generative model*, a.k.a. a simulator, which allows us to sample independently from the nominal controlled transition kernel $\overline{P}_{s,a}$, for any $(s, a) \in \mathcal{S} \times \mathcal{A}$. In particular, given sample size $n$, we sample i.i.d. $\{S^{(1)}_{s,a}, \cdots, S^{(n)}_{s,a}\}$ from $\overline{P}_{s,a}$ and construct the empirical transition probability

$$\overline{P}_{s,a,n}(s') := \frac{1}{n} \sum_{i=1}^n \mathbb{1}\left\{ S^{(i)}_{s,a} = s' \right\}. \tag{3.6}$$

Then, we define $\overline{P}_n := \{\overline{P}_{s,a,n} | (s, a) \in \mathcal{S} \times \mathcal{A}\}$ as the empirical nominal controlled transition kernel based on $n$ samples. We define the empirical uncertainty set $\mathcal{P}_n(f, \rho) := \times_{s \in \mathcal{S}} \mathcal{P}_{s,n}(f, \rho)$ where $\mathcal{P}_{s,n}(f, \rho)$ is from (3.2) by replacing $\overline{P}_{s,a}$ with $\overline{P}_{s,a,n}$. Again, the dependence on $(f, \rho)$ will be suppressed for simplicity.

Similarly, the empirical DR Bellman operator $\hat{\mathbf{T}}^\pi$ is defined as in (3.5) with $\mathcal{P}$ replaced by $\mathcal{P}_n$. The corresponding optimal empirical DR Bellman operator is $\hat{\mathbf{T}}^*(v)(s) := \sup_{\pi \in \Pi} \hat{\mathbf{T}}^\pi(v)(s), \forall s \in \mathcal{S}$.

Equipped with these definitions, we present our strategy to estimate the optimal value of the S-rectangular robust MDP via the empirical value function. This is motivated by the fact that $V^*_\mathcal{P} = v^*$ where $v^*$ solves (3.4).

**Definition 4** (Empirical Bellman Estimator). Given divergence function $f$ and radius parameter $\rho$, let $\mathcal{P} = \mathcal{P}(f, \rho)$ and $\mathcal{P}_n = \mathcal{P}_n(f, \rho)$. We define the empirical Bellman estimator $\hat{v}$ to $V^*_\mathcal{P}$ as the unique solution to the fixed point equation $\hat{\mathbf{T}}^*(\hat{v}) = \hat{v}$.

The rest of this paper is dedicated to theoretical analyses and numerical validation of the statistical efficiency of estimating $V^*_\mathcal{P} = v^*$ using $\hat{v}$. We conclude this section by introducing the following important proposition that provides an upper bound on the $l_\infty$ estimation error.

**Proposition 1.** *Let $v^*, \hat{v}$ be the solution of $\mathcal{T}^*(v) = v$ and $\hat{\mathbf{T}}^*(v) = v$, respectively. Then, the estimation error is upper bounded by*

$$\|\hat{v} - v^*\|_\infty \le \frac{1}{1 - \gamma} \left\| \hat{\mathbf{T}}^*(v^*) - \mathcal{T}^*(v^*) \right\|_\infty$$

*with probability 1.*

The proof of Proposition 1 is deferred to Appendix A.

## 4 Sample Complexity Bounds for the Empirical Bellman Estimator

In this section, we establish sample complexity upper bounds to achieve an absolute $\epsilon$ error in $l_\infty$ distance when estimating $V_{\mathcal{P}}^*$ using $\hat{v}$. We focus on two specific $f$-divergence uncertainty models. When $f_{\mathrm{KL}}(t) = f(t) = t\log t$, the corresponding uncertainty set $\mathcal{P}_s(f_{\mathrm{KL}}, \rho)$ is based on the Kullback–Leibler (KL) divergence, which is widely used in the machine learning literature. Alternatively, when $f = f_k$ as defined in Definition 6, the resulting $f_k$-divergence model captures another well-studied class of uncertainty sets [4].

We note that our analysis techniques are applicable to a broader class of smooth divergence functions $f$. However, we focus on these two representative cases for demonstration purposes. This reflects that achieving near-tight sample complexity bounds often requires leveraging specific structural properties of the divergence. In particular, we highlight the desirable feature that, in the regime where the radius $\rho \downarrow 0$, our bounds remain $O(1)$ in $\rho$, avoiding the diverging sample complexity upper bounds established in earlier results, as discussed in [25].

To facilitate our analysis and establish sample complexity results, we define the minimum support probability as a complexity metric parameter as follows.

**Definition 5.** Define the minimum support probability as

$$\mathfrak{p}_\wedge := \min_{s,a \in \mathcal{S} \times \mathcal{A}} \min_{s' \in \mathcal{S}: \overline{P}_{s,a}(s') > 0} \overline{P}_{s,a}(s')$$

As noted in the literature, the use of $\mathfrak{p}_\wedge$ as a complexity metric is well justified. In the KL case, the convergence rate of the estimation error can degrade arbitrarily, depending on the specific MDP instance, if there is no lower bound on the minimum support probability. In particular, the rate can be as slow as $\Omega(n^{-1/\beta})$ for any $\beta \geq 2$ as the sample size $n$ tends to infinity [19]. Similar negative results hold in the $f_k$-divergence setting when the parameter $k$ approaches 1 [3], highlighting the necessity of such a complexity measure.

### 4.1 The Kullback-Leibler Divergence Uncertainty Set

In this section, we present sample complexity results under the KL-divergence uncertainty set. Our analysis relies on the following dual representation of the DR Bellman operator and its empirical version.

**Lemma 1.** With $\mathcal{P} = \mathcal{P}(f_{\mathrm{KL}}, \rho)$ where $f_{\mathrm{KL}}(t) = t\log t$ and $\rho \in (0, \infty)$, for any $\pi \in \Pi$ and $s \in \mathcal{S}$, the dual form of the DR Bellman operator with KL uncertainty set $\mathcal{P}$ is

$$\mathcal{T}^\pi(v)(s) = \sup_{\lambda \geq 0} \left( -\lambda |\mathcal{A}| \rho - \sum_{a \in \mathcal{A}} \lambda \log \mathbb{E}_{\overline{P}_{s,a}} \left[ \exp\left( -\frac{\pi(a|s)(R(s,a,S) + \gamma v(S))}{\lambda} \right) \right] \right). \quad (4.1)$$

The KL empirical DR Bellman operator $\hat{\mathbf{T}}^\pi$ satisfies (4.1) with $\overline{P}_{s,a}$ replaced by $\overline{P}_{s,a,n}$.

The proof of Lemma 1 is provided in Appendix B.1. Building on this dual formulation, we next analyze the statistical error between the empirical and population DR Bellman operators.

**Proposition 2.** Under the KL-divergence uncertainty set with any $\rho \in (0, \infty)$, for any $v : \mathcal{S} \to \mathbb{R}$ and $n \geq 12\mathfrak{p}_\wedge^{-1} \log(4|\mathcal{S}|^2|\mathcal{A}|/\eta)$, with probability at least $1 - \eta$,

$$\|\hat{\mathbf{T}}^*(v) - \mathcal{T}^*(v)\|_\infty \leq \frac{9\|R + \gamma v\|_\infty}{\sqrt{n\mathfrak{p}_\wedge}} \sqrt{\log(4|\mathcal{S}|^2|\mathcal{A}|/\eta)}.$$

The proof of Proposition 2 is provided in Appendix C.1. Then, combining Proposition 2 with Proposition 1, and the fact that $\|R + \gamma v^*\|_\infty \leq 1/(1-\gamma)$ under our assumption that $R \in [0, 1]$, we arrive at the following theorem. The proof is presented in Appendix C.4.

**Theorem 1.** Under the KL-divergence uncertainty set with any $\rho \in (0, \infty)$ and $n \geq 12\mathfrak{p}_\wedge^{-1} \log(4|\mathcal{S}|^2|\mathcal{A}|/\eta)$, with probability at least $1 - \eta$,

$$\|\hat{v} - v^*\|_\infty \leq \frac{9}{(1-\gamma)^2 \sqrt{n\mathfrak{p}_\wedge}} \sqrt{\log(4|\mathcal{S}|^2|\mathcal{A}|/\eta)}.$$

**Remark 1.** Therefore, under the KL-divergence, to achieve an $\epsilon$ absolute error of estimating $V_{\mathcal{P}}^* = v^*$ with $\hat{v}$ in $l_\infty$ norm w.h.p., we need a total of $\widetilde{O}(|S||A|(1-\gamma)^{-4}\mathfrak{p}_\wedge^{-1}\epsilon^{-2})$ samples from the simulator.

## 4.2    $f_k$-Divergence Uncertainty Set

Next, we consider a subclass of the Cressie-Read family of $f_k$-divergence with $k \in (1, \infty)$, as studied in Duchi and Namkoong [4].

**Definition 6.** For $k \in (1, \infty)$, the $f_k$-divergence is defined by the divergence functions $f_k(t) := (t^k - kt + k - 1)/(k(k-1))$. We also define $k^* = k/(k-1)$.

Notably, when $k = 2$, the $f_2$-divergence is the $\chi^2$-divergence, which sees extensive application in the statistical testing literature. Moreover, when $k \downarrow 1$, the $f_k$ induced divergence converges to KL.

The analysis for $f_k$-divergence uncertainty sets follows the same strategy to KL-divergence in the previous subsection. Below we summarise the main results.

**Lemma 2.** *With $\mathcal{P} = \mathcal{P}(f_k, \rho)$ and $\rho \in (0, \infty)$, for any $\pi \in \Pi$ and $s \in \mathcal{S}$, the dual form of the DR Bellman operator with $f_k$ uncertainty set $\mathcal{P}$ is*

$$\mathcal{T}^\pi(v)(s) = -\sup_{\eta \in \mathbb{R}^{|\mathcal{A}|}} \left[ c \left( \sum_{a \in \mathcal{A}} \mathbb{E}_{\overline{P}_{s,a}} \left[ (\eta_a - \pi(a|s)[R(s,a,S) + \gamma v(S)])_+^{k^*} \right] \right)^{\frac{1}{k^*}} + \sum_{a \in \mathcal{A}} \eta_a \right]$$

*where $c = c(k, \rho, |\mathcal{A}|) = |\mathcal{A}|^{1/k} (k(k-1)\rho + 1)^{1/k}$ and $(\cdot)_+ = \max(\cdot, 0)$. The $f_k$ empirical DR Bellman operator $\hat{\mathbf{T}}^\pi$ satisfies a similar equality with $\overline{P}_{s,a}$ replaced by $\overline{P}_{s,a,n}$. $\overline{P}_{s,a,n}$.*

The proof of Lemma 2 is provided in Appendix B.2. Again, with this dual representation of the DR Bellman operators and refined estimation error analysis, we arrive at the following result.

**Proposition 3.** *Under the $f_k$-divergence uncertainty set with any $\rho \in (0, \infty)$, for any $v : \mathcal{S} \to \mathbb{R}$ and $n \geq 12\mathfrak{p}_\wedge^{-1} \log(4|\mathcal{S}|^2|\mathcal{A}|/\eta)$, with probability at least $1 - \eta$,*

$$\|\hat{\mathbf{T}}^*(v) - \mathcal{T}^*(v)\|_\infty \leq \frac{3 \cdot 2^{k^*} k^* \|R + \gamma v\|_\infty}{\sqrt{n\mathfrak{p}_\wedge}} \sqrt{\log\left(4|\mathcal{S}|^2|\mathcal{A}|/\eta\right)}.$$

The proof of Proposition 3 is provided in Appendix D.1. This, combined with Proposition 1, implies the following error convergence bound, whose proof is deferred to Appendix D.4.

**Theorem 2.** *Under the $f_k$-divergence uncertainty set with any $\rho \in (0, \infty)$ and $n \geq 12\mathfrak{p}_\wedge^{-1} \log(4|\mathcal{S}|^2|\mathcal{A}|/\eta)$, with probability at least $1 - \eta$,*

$$\|\hat{v} - v^*\|_\infty \leq \frac{3 \cdot 2^{k^*} k^*}{(1-\gamma)^2 \sqrt{n\mathfrak{p}_\wedge}} \sqrt{\log(4|\mathcal{S}|^2|\mathcal{A}|/\eta)}.$$

**Remark 2.** Therefore, under the $f_k$-divergence for a fixed $k$, to achieve an $\epsilon$ absolute error of estimating $V_{\mathcal{P}}^* = v^*$ with $\hat{v}$ in $l_\infty$ norm w.h.p., we also need a total of $\widetilde{O}(|S||A|(1-\gamma)^{-4}\mathfrak{p}_\wedge^{-1}\epsilon^{-2})$ samples from the simulator.

# 5    Numerical Experiments

In this section, we present two sets of numerical examples. In Section 5.1, we revisit the robust inventory problem from Ho et al. [7], which features uncertain demand, to demonstrate the $n^{-1/2}$ error decay rate. In Section 5.2, we consider an example from Yang et al. [30] to illustrate the linear dependence on $|\mathcal{S}||\mathcal{A}|$, which matches the lower bound established in Yang et al. [30].

## 5.1    Robust Inventory Control Problems

We investigate the dependency of the estimation error $\varepsilon$ on the sample size $n$ and evaluate our approach on a classical discrete-time inventory management problem with stochastic demand and backlog [7]. In each period $t$, an agent decides the order quantity to maximize cumulative discounted rewards, accounting for holding costs, backlog penalties, and profits.

Let $I$ denote the maximum inventory level, $B$ the maximum backlog, and $O$ the maximum order quantity per period. The state space is defined as $\mathcal{S} = \{-B, \cdots, 0, \cdots, I\}$, the action space is

$\mathcal{A} = \{0, \cdots, O\}$, where $s_t \in \mathcal{S}$ and $a_t \in \mathcal{A}$ denote the inventory and order item at the beginning of period $t$. Demand $D_t \in \{0, \cdots, D_{\max}\}$ is an i.i.d. sequence, with distribution $P_D \in \Delta^{D_{\max}+1}$.

The MDP dynamics proceed as follows. Due to storage constraints, the effective order size is $\tilde{A}_t = \min(A_t, I - S_t)$. Then, the next state evolves as $S_{t+1} = \max(S_t + \tilde{A}_t - D_t, -B)$, ensuring that the backlog does not exceed $B$. The actual sales in period $t$ are given by $X_t = S_t - S_{t+1} + \tilde{A}_t$. A one-step reward $R(S_t, A_t, S_{t+1}) = pX_t + b\min(S_{t+1}, 0) - h\max(S_{t+1}, 0) - c\tilde{A}_t$ is collected, where $p$ is the sales price, $c$ is the purchase cost, $h$ is the holding cost, and $b$ is the penalty of backlog.

For our experiments, we set the parameters as follows: $I = 10$, $B = 5$, $O = 5$, $p = 3$, $c = 2$, $h = 0.2$, $b = 3$, $\gamma = 0.9$, and use the nominal demand distribution $P_D = [0.1, 0.2, 0.3, 0.3, 0.1]$, supported on $0, 1, 2, 3, 4$. For each $(s, a)$, we sample $n_0$ samples from the nominal transition kernels to generate the estimated transition probability $P_n$, and solve the DR-MDP problem with uncertainty size $\rho = 1/6$ using the algorithm presented in [7].

Figure 1 illustrates the relationship between the sample size $n$ and the error $\varepsilon$ between the empirical and population value functions. As shown in the log-log plot, the slope is approximately $-0.5$ for both the KL and $\chi^2$ cases, indicating that the error decreases at a rate proportional to $1/\sqrt{n}$.

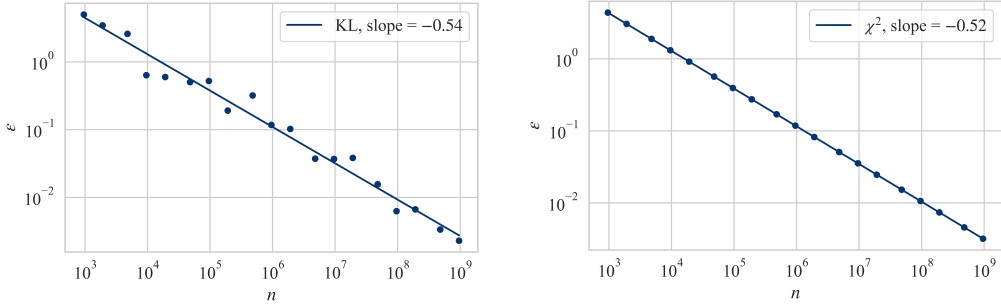

(a) Uncertainty sets based on KL-divergence      (b) Uncertainty sets based on $\chi^2$-divergence

Figure 1: Estimation error versus sample size $n$ in the robust inventory control problem.

## 5.2 MDP Instances from the Lower Bound Construction in Yang et al. [30]

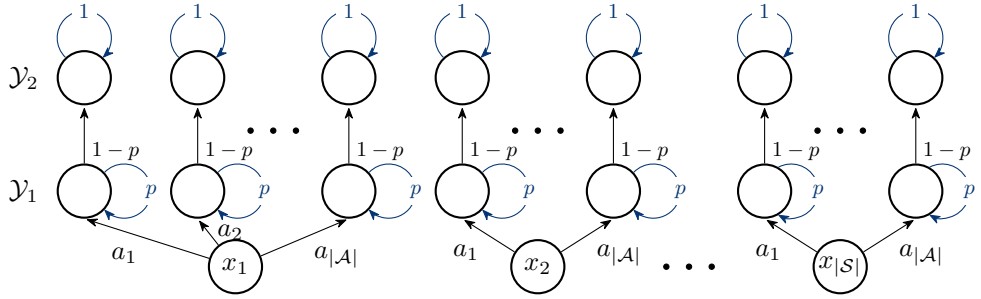

Figure 2: MDP instances from the lower bound construction in Yang et al. [30].

In this section, we investigate the relationship between the estimation error and the sizes of the state space $|\mathcal{S}|$ and action space $|\mathcal{A}|$. We adopt the classic MDP structure introduced in Gheshlaghi Azar et al. [5] and Yang et al. [30], which comprises three subsets: $\mathcal{S}$, $\mathcal{Y}_1$, and $\mathcal{Y}_2$, as illustrated in Figure 2.

Specifically, $\mathcal{S}$ denotes the set of all initial states, each associated with an action set $\mathcal{A}$. When an action $a_i \in \mathcal{A}$ is taken in state $s \in \mathcal{S}$, the system deterministically transitions (with probability 1) to the corresponding state $y_{1,s,a} \in \mathcal{Y}_1$. From each $y_{1,s,a}$, the system either remains in the same state with nominal probability $p$, or transitions to the corresponding absorbing state $y_{2,s,a} \in \mathcal{Y}_2$ with nominal probability $1 - p$. All states in $\mathcal{Y}_2$ are absorbing, meaning that once the system enters one of these states, it remains there indefinitely via a self-loop with probability 1. The reward function is

defined such that a reward of 1 is obtained only when the system is in any state within $\mathcal{Y}_1$; all other states yield a reward of 0. We solve the DR-MDP problem with uncertainty size $\rho = 0.1$.

In our experiments, we first fix $|\mathcal{A}| = 65$ and vary the number of states from 10 to 1000, with the results shown in Figure 3. We then fix $|\mathcal{S}| = 65$ and vary $|\mathcal{A}|$ over the same range, with the corresponding results presented in Figure 4.

To align with our theoretical results, we normalize the estimation error by dividing it by $\log(|\mathcal{S}||\mathcal{A}|)$. Figures 3 and 4 display the behavior of this normalized error as $|\mathcal{S}|$ and $|\mathcal{A}|$ vary, respectively. Specifically, for each $(s, a)$ pair, we use $n_0$ samples, resulting in a total of $n_0|\mathcal{S}||\mathcal{A}|$ samples. The left subfigures correspond to the KL-divergence case, while the right subfigures correspond to the $\chi^2$-divergence case. We observe that as either $|\mathcal{S}|$ or $|\mathcal{A}|$ increases, the normalized error is non-increasing. This is consistent with our theoretical analysis, which predicts that the sample complexity scales linearly (up to logarithmic factors) with the product $|\mathcal{S}||\mathcal{A}|$.

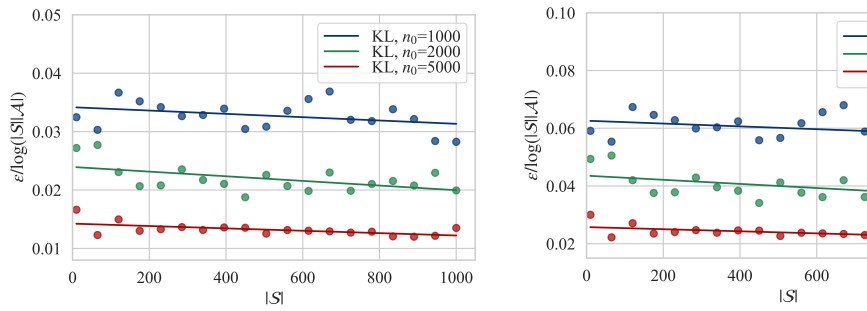

(a) Uncertainty sets based on KL-divergence  (b) Uncertainty sets based on $\chi^2$-divergence

Figure 3: Estimation error versus the number of states $|\mathcal{S}|$ for the MDP instances based on the lower bound construction in Yang et al. [30].

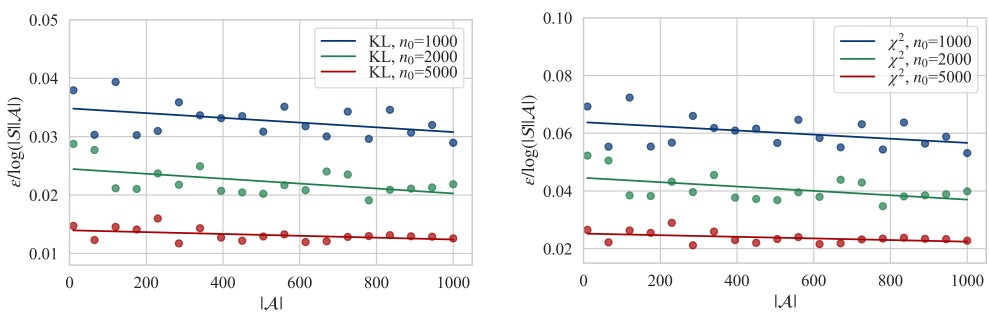

(a) Uncertainty sets based on KL-divergence  (b) Uncertainty sets based on $\chi^2$-divergence

Figure 4: Estimation error versus the number of states $|\mathcal{A}|$ for the MDP instances based on the lower bound construction in Yang et al. [30].

# 6 Conclusion and Future Work

In this paper, we present near-optimal sample complexity results for divergence-based S-rectangular robust MDPs in the discounted reward setting. Our results are the first to achieve optimal dependence on $|\mathcal{S}|$, $|\mathcal{A}|$, and $\varepsilon$ simultaneously. We acknowledge, however, two limitations: the reliance on access to a generative model and the presence of a gap between our upper bound and the minimax lower bound established in Yang [29]. As part of future work, we aim to develop provable theoretical guarantees for other settings, including model-free algorithms and offline reinforcement learning.

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

## A  Proofs of Value Function Error Bounds

In this section, we prove Proposition 1. We first show that both the population and the empirical S-rectangular Bellman operators $\mathcal{T}^*$ and $\hat{\mathbf{T}}^*$ are $\gamma$-contractions. This is a well-known fact, see for example [24]. We include a proof to make the paper self-contained.

**Lemma 3.** $\mathcal{T}^*$ and $\hat{\mathbf{T}}^*$ are $\gamma$-contraction operators on $(\mathcal{S} \to \mathbb{R}, \|\cdot\|_\infty)$; i.e. for all $v_1, v_2 : \mathcal{S} \to \mathbb{R}$,

$$\|\mathcal{T}^*(v_1) - \mathcal{T}^*(v_2)\|_\infty \le \gamma \|v_1 - v_2\|_\infty,$$
$$\|\hat{\mathbf{T}}^*(v_1) - \hat{\mathbf{T}}^*(v_2)\|_\infty \le \gamma \|v_1 - v_2\|_\infty.$$

*Proof.* Let

$$f(v)(s) = \sum_{a \in \mathcal{A}} \pi(a|s) \left[ \sum_{s' \in \mathcal{S}} P_{s,a}(s') \left( R(s,a,s') + \gamma v(s') \right) \right]$$

By definition, we have

$$|\mathcal{T}^*(v_1)(s) - \mathcal{T}^*(v_2)(s)| = \left| \sup_{\pi \in \Pi} \mathcal{T}^\pi(v_1)(s) - \sup_{\pi \in \Pi} \mathcal{T}^\pi(v_2)(s) \right|$$
$$= \left| \sup_{\pi \in \Pi} \inf_{P \in \mathcal{P}} f(v_1)(s) - \sup_{\pi \in \Pi} \inf_{P \in \mathcal{P}} f(v_2)(s) \right|.$$

Since $|\sup_X f - \sup_X g| \le \sup_X |f - g|$ and $|\inf_X f - \inf_X g| \le \sup_X |f - g|$, we have

$$|\mathcal{T}^*(v_1)(s) - \mathcal{T}^*(v_2)(s)|$$
$$\le \sup_{\pi \in \Pi} \sup_{P \in \mathcal{P}} |f(v_1)(s) - f(v_2)(s)|$$
$$= \sup_{\pi \in \Pi} \sup_{P \in \mathcal{P}} \left| \sum_{a,s'} \pi(a|s) P_{s,a}(s') R(s,a,s') + \gamma \sum_{a,s'} \pi(a|s) P_{s,a}(s') v_1(s') \right.$$
$$\left. - \sum_{a,s'} \pi(a|s) P_{s,a}(s') R(s,a,s') - \gamma \sum_{a,s'} \pi(a|s) P_{s,a}(s') v_2(s') \right|$$
$$= \sup_{\pi \in \Pi} \sup_{P \in \mathcal{P}} \gamma \left| \sum_{a,s'} \pi(a|s) P_{s,a}(s') v_1(s') - \sum_{a,s'} \pi(a|s) P_{s,a}(s') v_2(s') \right|$$
$$\le \gamma \|\pi_s\|_\infty \|P_s\|_\infty \|v_1 - v_2\|_\infty$$
$$= \gamma \|v_1 - v_2\|_\infty$$

where $\|P_s\|_\infty = \sup_{\|v\|_\infty = 1} \|P_s v\|_\infty$ is the induced operator norm. The above inequality holds for any $s \in \mathcal{S}$, which lead to

$$\|\mathcal{T}^*(v_1) - \mathcal{T}^*(v_2)\|_\infty \le \gamma \|v_1 - v_2\|_\infty.$$

We replace $P_s$ with $P_{s,n}$, notice that $\|P_{s,n}\|_\infty \le 1$

$$|\hat{\mathbf{T}}^*(v_1)(s) - \hat{\mathbf{T}}^*(v_2)(s)| \le \gamma \|\pi_s\|_\infty \|P_{s,n}\|_\infty \|v_1 - v_2\|_\infty$$
$$= \gamma \|v_1 - v_2\|_\infty.$$

which lead to

$$\|\hat{\mathbf{T}}^*(v_1) - \hat{\mathbf{T}}^*(v_2)\|_\infty \le \gamma \|v_1 - v_2\|_\infty.$$

$\square$

### A.1  Proof of Proposition 1

*Proof.* The proof of Proposition 1 follows a similar argument to that used for the continuous-case operator in [23].

440 Let $v_0 \equiv 0$ and $v_{k+1} = \hat{\mathbf{T}}^*(v_k)$. $\hat{u}$ is defined as the fix point of $\hat{\mathbf{T}}^*$ $\hat{v} = \hat{\mathbf{T}}^*(\hat{v})$

$$
\begin{aligned}
\Delta_{k+1} &= v_{k+1} - v^* \\
&= \hat{\mathbf{T}}^*(v_k) - \hat{\mathbf{T}}^*(v^*) + \hat{\mathbf{T}}^*(v^*) - \mathcal{T}^*(v^*) \\
&= \left[ \hat{\mathbf{T}}^*(v^* + \Delta_k) - \hat{\mathbf{T}}^*(v^*) \right] + \left[ \hat{\mathbf{T}}^*(v^*) - \mathcal{T}^*(v^*) \right] \\
&:= \mathbf{H}(\Delta_k) + V
\end{aligned}
$$

441 By Lemma 3, we have

$$
\|\mathbf{H}(\Delta_1) - \mathbf{H}(\Delta_2)\|_\infty = \left\| \hat{\mathbf{T}}^*(v^* + \Delta_1) - \hat{\mathbf{T}}^*(v^* + \Delta_2) \right\|_\infty \leq \gamma \|\Delta_1 - \Delta_2\|_\infty,
$$

442 therefore, $\mathbf{H}$ is also a $\gamma$-contraction operator. Then we show

$$
\|\Delta_k\|_\infty \leq \frac{\gamma^{k-1}}{1-\gamma} + \sum_{j=0}^{k-1} \gamma^j \|V\|_\infty
$$

443 by induction: for $k = 1$,

$$
\begin{aligned}
\|\Delta_1\|_\infty &\leq \|\mathbf{H}(\Delta_0)\|_\infty + \|V\|_\infty \\
&= \|\mathbf{H}(\Delta_0) - \mathbf{H}(0)\|_\infty + \|V\|_\infty \\
&\leq \gamma \|v^*\|_\infty + \|V\|_\infty \\
&\leq \frac{\gamma}{1-\gamma} + \|V\|_\infty .
\end{aligned}
$$

444 For any $k$, we have

$$
\begin{aligned}
\|\Delta_{k+1}\|_\infty &\leq \|\mathbf{H}(\Delta_k)\|_\infty + \|V\|_\infty \\
&= \|\mathbf{H}(\Delta_k) - \mathbf{H}(0)\|_\infty + \|V\|_\infty \\
&\leq \gamma \|\Delta_k\|_\infty + \|V\|_\infty \\
&\leq \gamma \left( \frac{\gamma^{k-1}}{1-\gamma} + \sum_{j=0}^{k-1} \|V\|_\infty \right) + \|V\|_\infty \\
&= \frac{\gamma^k}{1-\gamma} + \sum_{j=0}^{k} \|V\|_\infty.
\end{aligned}
$$

445 Therefore,

$$
\|\hat{v} - v^*\|_\infty = \lim_{k \to \infty} \|\Delta_k\|_\infty \leq \sum_{j=0}^{\infty} \gamma^j \|V\|_\infty = \frac{1}{1-\gamma} \left\| \hat{\mathbf{T}}^*(v^*) - \mathcal{T}^*(v^*) \right\|.
$$

446 $\qquad\qquad\qquad\qquad\qquad\qquad\qquad\qquad\qquad\qquad\qquad\qquad\qquad\qquad\qquad\qquad\qquad\qquad$ $\square$

# B  Strong Duality for Divergence-Based S-Rectangular Bellman Operators

448 The proofs for all $f$-divergence-based uncertainty sets follow a unified framework. We first present
449 Lemma 4, which gives a general dual formulation for any convex $f$-divergence. For the KL-
450 divergence and the $f_k$-divergence, we specialise this result by substituting the corresponding conjugate
451 functions $f^*$. The detailed derivations for the KL-divergence and the $f_k$-divergence are provided in
452 Appendix B.1 and B.2, respectively.

453 **Lemma 4.** *For any $f$-divergence uncertainty set, where $f : \mathbb{R}_+ \to \mathbb{R}$ is a convex function and*
454 $f(0) = 1$ *and satisfies* $f(0) = \lim_{t \downarrow 0} f(t)$, *the convex optimization problem*

$$
\inf_{P \in \mathcal{P}} \sum_{a \in \mathcal{A}} \pi(a|s) \mathbb{E}_{P_{s,a}} \left[ R(s, a, S) + \gamma v(S) \right]
$$

455 *can be reformulated as:*

$$
\sup_{\lambda \geq 0, \boldsymbol{\eta} \in \mathbb{R}^{|\mathcal{A}|}} -\lambda \sum_{a \in \mathcal{A}} \mathbb{E}_{\overline{P}_{s,a}} \left[ f^* \left( \frac{\eta_a - \pi(a|s) \left( R(s, a, S) + \gamma v(S) \right)}{\lambda} \right) \right] - \lambda |\mathcal{A}| \rho + \sum_{a \in \mathcal{A}} \eta_a
$$

456 *where* $f^*(t) = - \inf_{s \geq 0} \left( f(s) - st \right)$.

457 *Proof.* We follow the proof of Lemma 8.5 in [30], however, in our case, $R$ is determined by the next
458 state. We do a change of variables, let $L_{s,a}(s') = \frac{P_{s,a}(s')}{\overline{P}_{s,a}(s')}$. The original optimization problem can be
459 reformulated as:

$$\inf_{L_s \geq 0} \sum_{a \in \mathcal{A}} \pi(a|s) \mathbb{E}_{\overline{P}_{s,a}} \left[ L_{s,a} \left( R(s,a,S) + \gamma v(S) \right) \right]$$

$$\text{s.t.} \sum_{a \in \mathcal{A}} \mathbb{E}_{\overline{P}_{s,a}} [f(L_{s,a})] \leq |\mathcal{A}|\rho$$

$$\mathbb{E}_{\overline{P}_{s,a}} [L_{s,a}] = 1 \quad \text{for all } a \in \mathcal{A}$$

460 The Lagrange function of primal problem is

$$\mathcal{L}(L, \lambda, \boldsymbol{\eta}) = \sum_{a \in \mathcal{A}} \pi(a|s) \mathbb{E}_{\overline{P}_{s,a}} \left[ L_{s,a} \left( R(s,a,S) + \gamma v(S) \right) \right]$$

$$+ \lambda \left( \sum_{a \in \mathcal{A}} \mathbb{E}_{\overline{P}_{s,a}} [f(L_{s,a})] - |\mathcal{A}|\rho \right) - \sum_{a \in \mathcal{A}} \eta_a \left( \mathbb{E}_{\overline{P}_{s,a}} [L_{s,a}] - 1 \right)$$

461 Denoting $f^*(t) = -\inf_{s \geq 0} (f(s) - st)$,

$$\inf_{L_s \geq 0} \mathcal{L}(L, \lambda, \boldsymbol{\eta})$$

$$= \inf_{L_s \geq 0} \left( \sum_{a \in \mathcal{A}} \mathbb{E}_{\overline{P}_{s,a}} \left[ \pi(a|s) L_{s,a} \left( R(s,a,S) + \gamma v(S) \right) + \lambda f(L_{s,a}) - \eta_a L_{s,a} \right] \right) - \lambda |\mathcal{A}|\rho + \sum_{a \in \mathcal{A}} \eta_a$$

$$= \lambda \sum_{a \in \mathcal{A}} \inf_{L_{s,a} \geq 0} \mathbb{E}_{\overline{P}_{s,a}} \left[ \frac{\pi(a|s) \left( R(s,a,S) + \gamma v(S) \right) - \eta_a}{\lambda} L_{s,a} + f(L_{s,a}) \right] - \lambda |\mathcal{A}|\rho + \sum_{a \in \mathcal{A}} \eta_a$$

$$= -\lambda \sum_{a \in \mathcal{A}} \mathbb{E}_{\overline{P}_{s,a}} \left[ f^* \left( \frac{\eta_a - \pi(a|s) \left( R(s,a,S) + \gamma v(S) \right)}{\lambda} \right) \right] - \lambda |\mathcal{A}|\rho + \sum_{a \in \mathcal{A}} \eta_a$$

462 $\square$

## B.1 Proof of Lemma 1

464 *Proof.* Recall that for the KL-divergence, $f(t) = t \log t$, whose conjugate function $f^*(s) = e^{s-1}$.
465 Substituting $f^*$ into Lemma 4, we obtain the following dual form:

$$\sup_{\lambda \geq 0, \boldsymbol{\eta} \in \mathbb{R}^{|\mathcal{A}|}} -\lambda \sum_{a \in \mathcal{A}} \mathbb{E}_{\overline{P}_{s,a}} \left[ f^* \left( \frac{\eta_a - \pi(a|s) \left( R(s,a,S) + \gamma v(S) \right)}{\lambda} \right) \right] - \lambda |\mathcal{A}|\rho + \sum_{a \in \mathcal{A}} \eta_a$$

$$= \sup_{\lambda \geq 0, \boldsymbol{\eta} \in \mathbb{R}^{|\mathcal{A}|}} -\lambda \sum_{a \in \mathcal{A}} \exp\left( \frac{\eta_a - \lambda}{\lambda} \right) \mathbb{E}_{\overline{P}_{s,a}} \left[ \exp\left( \frac{-\pi(a|s) \left( R(s,a,S) + \gamma v(S) \right)}{\lambda} \right) \right]$$

$$- \lambda |\mathcal{A}|\rho + \sum_{a \in \mathcal{A}} \eta_a.$$

(B.1)

466 We first note that for each action $a$, the term $\lambda \mathbb{E}_{\overline{P}_{s,a}} [\cdot]$ is a positive constant with respect to $\eta_a$, while
467 the term $-\lambda \mathbb{E}_{\overline{P}_{s,a}} [\cdot] \exp((\eta_a - \lambda)/\lambda)$ is concave in $\eta_a$, since for any $c > 0$, the function $-c \exp(x)$
468 is concave. Moreover, the term $\sum_a \eta_a$ is affine, and hence concave. As the sum of concave functions
469 is concave, we conclude that (B.1) is concave in $\boldsymbol{\eta}$. Next, we optimize with respect to $\boldsymbol{\eta}$ by setting
470 the gradient with respect to each $\eta_a$ to zero:

$$- \exp\left( \frac{\eta_a - \lambda}{\lambda} \right) \mathbb{E}_{\overline{P}_{s,a}} \left[ \exp\left( \frac{-\pi(a|s) \left( R(s,a,S) + \gamma v(S) \right)}{\lambda} \right) \right] + 1 = 0$$

471 Solving for $\eta_a$, we obtain

$$\eta_a = \lambda - \lambda \log \mathbb{E}_{\overline{P}_{s,a}} \left[ \exp\left( \frac{-\pi(a|s) \left( R(s,a,S) + \gamma v(S) \right)}{\lambda} \right) \right].$$

(B.2)

Substituting (B.2) into (B.1), we obtain

$$\sup_{\lambda \geq 0, \boldsymbol{\eta} \in \mathbb{R}^{|\mathcal{A}|}} -\lambda \sum_{a \in \mathcal{A}} \mathbb{E}_{\overline{P}_{s,a}} \left[ \exp\left( \frac{-\pi(a|s)\left(R(s,a,S) + \gamma v(S)\right)}{\lambda} \right) \right] - \lambda |\mathcal{A}| \rho.$$

$\square$

## B.2 Proof of Lemma 2

*Proof.* We first introduce the conjugate function of $f_k$, which will be instrumental for deriving the dual representation of DR Bellman operator.

**Lemma 5** (Duchi and Namkoong [4], Section 2). *Recall that in $f_k$-divergence,*

$$f_k(t) := \frac{t^k - kt + k - 1}{k(k-1)}$$

*The conjugate function $f_k^*(s) = \sup_{t \geq 0}(st - f_k(t))$ is given by*

$$f_k^*(s) := \frac{1}{k} \left[ ((k-1)s + 1)_+^{k_*} - 1 \right]$$

*where $(x)_+ = \max(x, 0)$.*

Substituting $f_k^*$ into Lemma 4, and let $w_{s,a}(S) := \pi(a|s)\left(R(s,a,S) + \gamma v(S)\right)$, we obtain

$$\sup_{\lambda \geq 0, \eta \in \mathbb{R}^{|\mathcal{A}|}} -\sum_{a \in \mathcal{A}} \lambda \mathbb{E}_{\overline{P}_{s,a}} \left[ f^* \left( \frac{\eta_a - w_{s,a}(S)}{\lambda} \right) \right] - \lambda |\mathcal{A}| \rho + \sum_{a \in \mathcal{A}} \eta_a$$

$$= \sup_{\lambda \geq 0, \eta \in \mathcal{A}} -\sum_{a \in \mathcal{A}} \lambda \mathbb{E}_{\overline{P}_{s,a}} \left[ \frac{1}{k} \left[ \left( (k-1)\frac{\eta_a - w_{s,a}(S)}{\lambda} + 1 \right)_+^{k_*} - 1 \right] \right]$$

Since $k - 1 > 0$ and $\lambda > 0$ are constants with respect to the random variable $S$, we can factor them out of the expectation and the positive-part operator $(\cdot)_+$.

$$= \sup_{\lambda \geq 0, \eta \in \mathbb{R}^{|\mathcal{A}|}} -\frac{(k-1)^{k^*}}{k\lambda^{k^*-1}} \sum_{a \in \mathcal{A}} \mathbb{E}_{\overline{P}_{s,a}} \left[ \left( \eta_a - w_{s,a}(S) + \frac{\lambda}{k-1} \right)_+^{k^*} \right] - \lambda |\mathcal{A}| \left( \rho - \frac{1}{k} \right) + \sum_{a \in \mathcal{A}} \eta_a$$

Finally, we perform the change of variables, let $\tilde{\eta}_a = \eta_a + \frac{\lambda}{k-1}$, we obtain

$$= \sup_{\lambda \geq 0, \tilde{\eta} \in \mathbb{R}^{|\mathcal{A}|}} -\frac{(k-1)^{k^*}}{k\lambda^{k^*-1}} \sum_{a \in \mathcal{A}} \mathbb{E}_{\overline{P}_{s,a}} \left[ (\tilde{\eta}_a - w_{s,a}(S))_+^{k^*} \right] - \lambda |\mathcal{A}| \left( \rho + \frac{1}{k(k-1)} \right) + \sum_{a \in \mathcal{A}} \tilde{\eta}_a$$
(B.3)

Since $-\lambda^{-\alpha}$ is concave in $\lambda$ for any $\alpha > 0$, and $\lambda |\mathcal{A}| \left( \rho + \frac{1}{k(k-1)} \right)$ is an affine function of $\lambda$, it follows that (B.3) is concave with respect to $\lambda$. To optimize over $\lambda$, we take the derivative with respect to $\lambda$ and set it to zero, which yields:

$$\frac{(k-1)^{k^*}}{k(k-1)\lambda^{k^*}} \sum_{a \in \mathcal{A}} \mathbb{E}_{\overline{P}_{s,a}} \left[ (\tilde{\eta}_a - w_{s,a}(S))_+^{k^*} \right] - |\mathcal{A}| \left( \rho + \frac{1}{k(k-1)} \right) = 0$$

Multiply $k(k-1)$ on both side of the equation, we have

$$\frac{(k-1)^{k^*}}{\lambda^{k^*}} \sum_{a \in \mathcal{A}} \mathbb{E}_{\overline{P}_{s,a}} \left[ (\tilde{\eta}_a - w_{s,a}(S))_+^{k^*} \right] - |\mathcal{A}| \left( k(k-1)\rho + 1 \right) = 0$$

Therefore, we obtain

$$\lambda^* = (k-1)|\mathcal{A}|^{-1/k^*} \left( k(k-1)\rho + 1 \right)^{-1/k^*} \left( \sum_{a \in \mathcal{A}} \mathbb{E}_{\overline{P}_{s,a}} \left[ (\tilde{\eta}_a - w_{s,a}(S))_+^{k^*} \right] \right)^{1/k^*}$$

489   By substituting $\lambda^*$ into the equation (B.3) , we have

$$\sup_{\lambda \geq 0, \eta \in \mathbb{R}^{|\mathcal{A}|}} -\sum_{a \in \mathcal{A}} \lambda \mathbb{E}_{\overline{P}_{s,a}}\left[f^*\left(\frac{\eta_a - w_{s,a}(S)}{\lambda}\right)\right] - \lambda|\mathcal{A}|\rho + \sum_{a \in \mathcal{A}} \tilde{\eta}_a$$

$$= \sup_{\tilde{\eta} \in \mathbb{R}^{|\mathcal{A}|}} -\frac{k-1}{k}|\mathcal{A}|^{1/k}\left(k(k-1)\rho+1\right)^{1/k}\left(\sum_{a \in \mathcal{A}} \mathbb{E}_{\overline{P}_{s,a}}\left[(\tilde{\eta}_a - w_{s,a}(S))_+^{k^*}\right]\right)^{1/k^*}$$

$$-\frac{1}{k}|\mathcal{A}|^{1/k}\left(k(k-1)\rho+1\right)^{1/k}\left(\sum_{a \in \mathcal{A}} \mathbb{E}_{\overline{P}_{s,a}}\left[(\tilde{\eta}_a - w_{s,a}(S))_+^{k^*}\right]\right)^{1/k^*} + \sum_{a \in \mathcal{A}} \tilde{\eta}_a$$

$$= \sup_{\tilde{\eta} \in \mathbb{R}^{|\mathcal{A}|}} -|\mathcal{A}|^{1/k}\left(k(k-1)\rho+1\right)^{1/k}\left(\sum_{a \in \mathcal{A}} \mathbb{E}_{\overline{P}_{s,a}}\left[(\tilde{\eta}_a - w_{s,a}(S))_+^{k^*}\right]\right)^{1/k^*} + \sum_{a \in \mathcal{A}} \tilde{\eta}_a$$

490                                                                      $\square$

## C   Proofs of Properties of the Empirical Bellman Operator: KL Case

492   Our techniques in this section refine that in Wang et al. [25]. To follow the constructions in Wang
493   et al. [25], we introduce some notations. Consider $\mu_{s,a} \in \Delta(\mathcal{S})$ and its empirical version $\mu_{s,a,n}$
494   constructed from $n$ i.i.d samples from $\mu_{s,a}$. Define the collection of these measures under state $s$ as
495   $\boldsymbol{\mu}_s := \{\mu_{s,a} : a \in \mathcal{A}\}$. For a function $u : \mathcal{S} \to \mathbb{R}$ and for each $s \in \mathcal{S}$, we define:

$$\|u\|_{\infty, \boldsymbol{\mu}_s} = \max_{a \in \mathcal{A}} \|u\|_{L^\infty(\mu_{s,a})},$$

$$\left\|\frac{dm_n}{d\mu_n(t)}\right\|_{\infty, \boldsymbol{\mu}_s} = \max_{a \in \mathcal{A}} \left\|\frac{dm_{a,n}}{d\mu_{a,n}(t)}\right\|_{L^\infty(\mu_{s,a})}.$$

496   For the supremum over all states, we define

$$\|u\|_\infty = \sup_{s \in \mathcal{S}} \|u\|_{\infty, \boldsymbol{\mu}_s}.$$

497   We define a "good event" under which the empirical measure $\mu_{s,a,n}$ uniformly approximates the
498   population measure $\mu_{s,a}$ with relative error bounded by $\delta_0$ across all actions $a \in \mathcal{A}$. Formally, this
499   event is given by

$$\Omega_{n,\delta_0}(\boldsymbol{\mu}_s) = \left\{\omega : \sup_{a \in \mathcal{A}} \sup_{s' \in \mathcal{S}} \left|\frac{\mu_{s,a,n}(\omega)(s') - \mu_{s,a}(s')}{\mu_{s,a}(s')}\right| \leq \delta_0\right\}.$$

500   Further, the good event over all states is defined as

$$\Omega_{n,\delta_0} = \bigcap_{s \in \mathcal{S}} \Omega_{n,\delta_0}(\boldsymbol{\mu}_s) = \left\{\omega : \sup_{s \in \mathcal{S}} \sup_{a \in \mathcal{A}, s' \in \mathcal{S}} \left|\frac{\mu_{s,a,n}(\omega)(s') - \mu_{s,a}(s')}{\mu_{s,a}(s')}\right| \leq \delta_0\right\}.$$

501   For notation simplicity, we suppress the dependence on the state variable $s$. Consider a function
502   $u : \mathcal{S} \to \mathbb{R}$. The dual function under KL-divergence is given by:

$$f(\boldsymbol{\mu}, u, \lambda) := -\lambda|\mathcal{A}|\rho - \sum_{a \in \mathcal{A}} \lambda \log \mu_a\left[e^{-d_a u/\lambda}\right], \tag{C.1}$$

503   where $\lambda > 0$ is the dual regularization parameter, and we denote $d_a := \pi(a|s)$ for simplicity.

504   We define the deviation between empirical and true measures as

$$m_{a,n} = \mu_{a,n} - \mu_a,$$

505   and their convex interpolation by

$$\mu_{a,n}(t) = t\mu_a + (1-t)\mu_{a,n}.$$

## C.1 Proof of Proposition 2

*Proof.* By definition and $|\sup_X f - \sup_X g| \leq \sup_X |f - g|$, we have

$$P\left(\left|\hat{\mathbf{T}}^*(v)(s) - \mathcal{T}^*(v)(s)\right| > t\right)$$

$$= P\left(\left|\sup_{\pi \in \Pi} \hat{\mathbf{T}}^\pi(v)(s) - \sup_{\pi \in \Pi} \mathcal{T}^\pi(v)(s)\right| > t\right)$$

$$\leq P\left(\sup_{\pi \in \Pi} \left|\hat{\mathbf{T}}^\pi(v)(s) - \mathcal{T}^\pi(v)(s)\right| > t\right).$$

Using (C.1) to express Bellman operator, we obtain

$$\left|\hat{\mathbf{T}}^\pi(v)(s) - \mathcal{T}^\pi(v)(s)\right| \leq \left|\sup_{\lambda > 0} f(P_{s,n}, R(s, \cdot, \cdot) + \gamma v, \lambda) - \sup_{\lambda > 0} f(P_s, R(s, \cdot, \cdot) + v, \lambda)\right| \tag{C.2}$$
$$\leq \sup_{\lambda > 0} |f(P_s, R(s, \cdot, \cdot) + \gamma v, \lambda) - f(P_s, R(s, \cdot, \cdot) + v, \lambda)|.$$

We analyze the sensitivity of the mapping $\boldsymbol{\mu} \to f(\boldsymbol{\mu}, u, \lambda)$. For any fixed $u$, $\boldsymbol{\mu}$ and $\boldsymbol{\mu}_n$, define

$$g_n(t, \lambda) = f(\boldsymbol{\mu}_n(t), u, \lambda).$$

According to mean value theorem, there exists $\tau \in (0, 1)$ satisfies:

$$|f(\boldsymbol{\mu}_n, u, \lambda) - f(\boldsymbol{\mu}, u, \lambda)| = |g_n(0, \lambda) - g_n(1, \lambda)|$$
$$= \left|\partial_t g_n(t, \lambda)\Big|_{t=\tau}\right|$$
$$= \left|\sum_{a \in \mathcal{A}} \lambda \frac{m_{a,n}[e^{-d_a u/\lambda}]}{\mu_{a,n}(\tau)[e^{-d_a u/\lambda}]}\right|$$

To bound the difference above, we invoke the following lemma.

**Lemma 6.** *For any fixed $u$ and $\pi$, $\mu_n \ll \mu$, we have that*

$$\sup_{\lambda \geq 0} \left|\sum_{a \in \mathcal{A}} \lambda \frac{m_{a,n}[e^{-d_a u/\lambda}]}{\mu_{a,n}(t)[e^{-d_a u/\lambda}]}\right| \leq 2\|u\|_\infty \left\|\frac{dm_n}{d\mu_n(t)}\right\|_{\infty,\boldsymbol{\mu}}.$$

The proof is deferred to Appendix C.2. According to lemma 6, we have

$$\sup_{\lambda \geq 0} |f(\boldsymbol{\mu}_n, u, \lambda) - f(\boldsymbol{\mu}, u, \lambda)| \leq 2\|u\|_\infty \left\|\frac{dm_n}{d\mu_n(t)}\right\|_{\infty,\boldsymbol{\mu}}.$$

We decomposed the probability using the event $\Omega_{n,\delta_0}(\boldsymbol{\mu})$ where the empirical estimates are close to the population measures:

$$P\left(\sup_{\lambda \geq 0, d \in \Delta(\mathcal{A})} |f(\mu_n, u, \lambda) - f(\mu, u, \lambda)| > t\right)$$

$$\leq P(\Omega_{n,\delta_0}(\boldsymbol{\mu})^c) + P\left(2\|u\|_\infty \left\|\frac{dm_n}{d\mu_n(\tau)}\right\|_{\infty,\boldsymbol{\mu}} > t, \Omega_{n,\delta_0}(\boldsymbol{\mu})\right)$$

To control the denominator $\mu_{a,n}(\tau)(s')$ appearing in the bound, we use the following lemma, which asserts that under the good event, the empirical and population measures remain close for all $t \in [0, 1]$:

**Lemma 7.** *For any $s'$ with $\mu(s') > 0$, the measure $\mu_n(t)(s')$ satisfies*

$$(1 - \delta_0)\mu(s') \leq \mu_n(t)(s') \leq (1 + \delta_0)\mu(s'), \quad \forall t \in [0, 1].$$

521 The proof is deferred to Appendix C.3. By using lemma 7, we have $\mu_{a,n}(\tau)(s') \geq (1-\delta_0)\mu_a(s')$,
522 therefore,

$$\leq P\left(\sup_{a,s'}\left|\frac{\mu_{a,n}(s') - \mu_a(s')}{\mu_a(s')}\right| > \delta_0\right) + P\left(2\|u\|_\infty \sup_{a,s'}\left|\frac{\mu_{a,n}(s') - \mu_a(s')}{(1-\delta_0)\mu_a(s')}\right| > t\right).$$

523 By using the multiplicative Chernoff bound and Bernstein inequality, we have

$$\leq P\left(\sup_{a,s'}\left|\frac{1}{n}\sum_{i=1}^n \mathbb{1}(S_i = s') - \mu_a(s')\right| > \delta_0\mu_a(s')\right)$$

$$+ P\left(\frac{2}{1-\delta_0}\|u\|_\infty \sup_{a,s'}\frac{1}{\mu_a(s')}\left|\frac{1}{n}\sum_{i=1}^n \mathbb{1}(S_i = s') - \mu_a(s')\right| > t\right)$$

$$\leq 2\sum_{a\in\mathcal{A}}\sum_{s'\in\mathcal{S}}\left(\exp\left(-\frac{\delta_0^2 n\mu_a(s')}{3}\right) + \exp\left(-\frac{t^2}{2}\left(\frac{4\|u\|_\infty^2}{(1-\delta_0)^2 n\mu_a(s')} + \frac{2\|u\|_\infty t}{3(1-\delta_0)n\mu_a(s')}\right)^{-1}\right)\right).$$

524 Since $\mu_a(y) \geq \mathfrak{p}_\wedge$, and both exponential term above is monotonically decreasing over $\mu_a(s')$, we
525 have

$$\leq 2|\mathcal{A}||\mathcal{S}|\left(\exp\left(-\frac{\delta_0^2 n\mathfrak{p}_\wedge}{3}\right) + \exp\left(-\frac{t^2}{2}\left(\frac{4\|u\|_\infty^2}{(1-\delta_0)^2 n\mathfrak{p}_\wedge} + \frac{2\|u\|_\infty t}{3(1-\delta_0)n\mathfrak{p}_\wedge}\right)^{-1}\right)\right).$$

526 Recall from (C.2) that

$$P\left(\left|\hat{\mathbf{T}}^*(v)(s) - \mathcal{T}^*(v)(s)\right| > t\right) \leq P\left(\sup_{\lambda>0, d_a\in\Delta(\mathcal{A})}|f(P_s, R(s,\cdot,\cdot) + \gamma v, \lambda) - f(P_s, R(s,\cdot,\cdot) + \gamma v, \lambda)|\right)$$

527 Replacing $\boldsymbol{\mu}$ with $P_s$ and $\boldsymbol{\mu}_n$ with $P_{s,n}$ and choose $\delta_0 = \frac{1}{2}$, by union bound, we have

$$P\left(\left\|\hat{\mathbf{T}}^*(v) - \mathcal{T}^*(v)\right\|_\infty > t\right)$$

$$\leq P\left(\sup_s \sup_{\lambda\geq 0, d\in\Delta(\mathcal{A})}|f(P_{s,n}, R(s,\cdot,\cdot) + \gamma v, \lambda) - f(P_s, R(s,\cdot,\cdot) + v, \lambda)| > t\right)$$

$$\leq 2|\mathcal{S}|^2|\mathcal{A}|\exp\left(-\frac{n\mathfrak{p}_\wedge}{12}\right) + 2|\mathcal{S}|^2|\mathcal{A}|\exp\left(-\frac{t^2}{2\gamma^2}\left(\frac{16\|R(s,\cdot,\cdot) + \gamma v\|_\infty^2}{n\mathfrak{p}_\wedge} + \frac{4\|R(s,\cdot,\cdot) + \gamma v\|_\infty t}{3\gamma n\mathfrak{p}_\wedge}\right)^{-1}\right).$$

528 Set each term to be less than $\eta/2$, we need

$$n \geq \frac{12}{\mathfrak{p}_\wedge}\log\left(4|\mathcal{S}|^2|\mathcal{A}|/\eta\right) \tag{C.3}$$

$$t \geq \frac{8\|R + \gamma v\|_\infty}{3n\mathfrak{p}_\wedge}\log\left(4|\mathcal{S}|^2|\mathcal{A}|/\eta\right) + \frac{4\|R + \gamma v\|_\infty}{\sqrt{n\mathfrak{p}_\wedge}}\sqrt{2\log\left(4|\mathcal{S}|^2|\mathcal{A}|/\eta\right)}. \tag{C.4}$$

529 Under (C.3), we have

$$\frac{\log(4|\mathcal{S}|^2|\mathcal{A}|/\eta)}{n\mathfrak{p}_\wedge} \leq \sqrt{\frac{\log(4|\mathcal{S}|^2|\mathcal{A}|/\eta)}{n\mathfrak{p}_\wedge}}.$$

530 By substituting this bound into (C.4), we have

$$\frac{8\|R + \gamma v\|_\infty}{3n\mathfrak{p}_\wedge}\log\left(4|\mathcal{S}|^2|\mathcal{A}|/\eta\right) + \frac{4\|R + \gamma v\|_\infty}{\sqrt{n\mathfrak{p}_\wedge}}\sqrt{2\log\left(4|\mathcal{S}|^2|\mathcal{A}|/\eta\right)}$$

$$\leq \left(\frac{8}{3} + 4\sqrt{2}\right)\frac{\|R + \gamma v\|_\infty}{\sqrt{n\mathfrak{p}_\wedge}}\sqrt{\log\left(4|\mathcal{S}|^2|\mathcal{A}|/\eta\right)}$$

$$\leq \frac{9\|R + \gamma v\|_\infty}{\sqrt{n\mathfrak{p}_\wedge}}\sqrt{\log\left(4|\mathcal{S}|^2|\mathcal{A}|/\eta\right)}$$

531 Therefore, for when $n$ specifies (C.3) and $t$ satisfies

$$t \geq \frac{9\|R + \gamma v\|_\infty}{\sqrt{n\mathfrak{p}_\wedge}}\sqrt{\log\left(4|\mathcal{S}|^2|\mathcal{A}|/\eta\right)},$$

532 we have

$$P\left(\left\|\hat{\mathbf{T}}^*(v) - \mathcal{T}^*(v)\right\|_\infty > t\right) \leq \eta.$$

533 This implies Proposition 2. □

## C.2 Proof of Lemma 6

535 *Proof.* Observe that multiplying the numerator and denominator by $e^{d_a\|u\|_{L^\infty(\mu_a)}/\lambda}$ preserves the
536 value of the fraction. This is equivalent to rewriting the exponential terms as:

$$\left|\sum_{a\in\mathcal{A}}\lambda\frac{m_{a,n}[e^{-d_a u/\lambda}]}{\mu_{a,n}(t)[e^{-d_a u/\lambda}]}\right| = \left|\sum_{a\in\mathcal{A}}\lambda\frac{m_{a,n}[e^{d_a(\|u\|_{L^\infty(\mu_a)}-u)/\lambda}]}{\mu_{a,n}(t)[e^{d_a(\|u\|_{L^\infty(\mu_a)}-u)/\lambda}]}\right|.$$

537 Since $m_{a,n} = \mu_{a,n} - \mu_a$, for any constant $c$, we have $m_{a,n}[c] = 0$, which lead to

$$= \left|\sum_{a\in\mathcal{A}}\lambda\frac{m_{a,n}[e^{d_a(\|u\|_{L^\infty(\mu_a)}-u)/\lambda} - 1]}{\mu_{a,n}(t)[e^{d_a(\|u\|_{L^\infty(\mu_a)}-u)/\lambda}]}\right|.$$

538 For any measure $m, \mu$ and random variable $w_1, w_2$, the following equation holds:

$$\left|\frac{m[w_1]}{\mu[w_2]}\right| = \left|\frac{\sum_s m(s)w_1(s)}{\sum_s \mu(s)w_2(s)}\right|$$

$$= \left|\left(\sum_s \mu(s)\frac{m(s)}{\mu(s)}w_2(s)\frac{w_1(s)}{w_2(s)}\right)\left(\sum_s \mu(s)w_2(s)\right)^{-1}\right|$$

$$\leq \left|\frac{\sum_s \mu(s)w_2(s)}{\sum_s \mu(s)w_2(s)}\right|\cdot\max_s\left|\frac{m(s)}{\mu(s)}\right|\cdot\max_s\left|\frac{w_1(s)}{w_2(s)}\right|$$

$$= \left\|\frac{dm}{d\mu}\right\|_{L^\infty(\mu)}\left\|\frac{w_1}{w_2}\right\|_{L^\infty(\mu)}.$$

539 Applying this result and $\left|\sum\cdot\right| \leq \sum|\cdot|$, we obtain

$$\left|\sum_{a\in\mathcal{A}}\lambda\frac{m_{a,n}[e^{-d_a u/\lambda}]}{\mu_{a,n}(t)[e^{-d_a u/\lambda}]}\right| \leq \sum_{a\in\mathcal{A}}\left\|\lambda\frac{e^{d_a(\|u\|_{L^\infty(\mu_a)}-u)/\lambda} - 1}{e^{d_a(\|u\|_{L^\infty(\mu_a)}-u)/\lambda}}\right\|_{L^\infty(\mu_a)}\left\|\frac{dm_{a,n}}{d\mu_{a,n}(t)}\right\|_{L^\infty(\mu_a)}.$$

540 Notice that when $x > 0$, we have $e^x - 1 > xe^x$, then we obtain

$$\leq \sum_{a\in\mathcal{A}}\left\|\lambda\frac{\frac{d_a(\|u\|_{L^\infty(\mu_a)}-u)}{\lambda}e^{d_a(\|u\|_{L^\infty(\mu_a)}-u)/\lambda}}{e^{d_a(\|u\|_{L^\infty(\mu_a)}-u)/\lambda}}\right\|_{L^\infty(\mu_a)}\left\|\frac{dm_{a,n}}{d\mu_{a,n}(t)}\right\|_{L^\infty(\mu_a)}$$

$$\leq \sum_{a\in\mathcal{A}}\left\|d_a(\|u\|_{L^\infty(\mu_a)} - u)\right\|_{L^\infty(\mu_a)}\left\|\frac{dm_{a,n}}{d\mu_{a,n}(t)}\right\|_{L^\infty(\mu_a)}$$

$$\leq \sum_{a\in\mathcal{A}}2d_a\|u\|_\infty\left\|\frac{dm_{a,n}}{d\mu_{a,n}(t)}\right\|_{L^\infty(\mu_a)}$$

$$\leq 2\|u\|_\infty\left\|\frac{dm_n}{d\mu_n(t)}\right\|_{\infty,\boldsymbol{\mu}}$$

541 as claimed. □

### C.3 Proof of Lemma 7

*Proof.* On the event $\Omega_{n,p}$, the empirical measure satisfies $\sup_{s'\in\mathcal{S}}\left|\frac{\mu_n(s')-\mu(s')}{\mu(s')}\right| \leq \delta_0$. Hence, for any $s'$ with $\mu(s') > 0$, we have:

$$(1-\delta_0)\mu(s') \leq \mu_n(s') \leq (1+\delta_0)\mu(s').$$

Substituting in the above bound on $\mu_n(s')$ into the definition of $\mu_n(t)(s')$ gives

$$(1-(1-t)\delta_0)\mu(s') \leq t\mu(s') + (1-t)\mu_n(s') \leq (1+(1-t)\delta_0)\mu(s').$$

For all $t \in [0,1]$, $(1-t) \leq 1$, therefore, we have

$$(1-\delta_0)\mu(s') \leq \mu_n(t)(s') \leq (1+\delta_0)\mu(s').$$

$\square$

### C.4 Proof of Theorem 1

*Proof.* Substituting $\|R + \gamma v\|_\infty \leq 1/(1-\gamma)$ into the bound from Proposition 2 and applying Proposition 1, we obtain the stated result.

$$\begin{aligned}
\|\hat{v} - v^*\|_\infty &\leq \frac{1}{1-\gamma}\left\|\hat{\mathbf{T}}^*(v^*) - \mathcal{T}^*(v^*)\right\|_\infty \\
&\leq \frac{9\|R+\gamma v\|_\infty}{(1-\gamma)\sqrt{n\mathfrak{p}_\wedge}}\sqrt{\log\left(4|\mathcal{S}|^2|\mathcal{A}|/\eta\right)} \\
&\leq \frac{9}{(1-\gamma)^2\sqrt{n\mathfrak{p}_\wedge}}\sqrt{\log\left(4|\mathcal{S}|^2|\mathcal{A}|/\eta\right)}
\end{aligned}$$

with probability $1 - \eta$.

$\square$

## D Proofs of Properties of the Empirical Bellman Operator: $f$-Divergence Case

### D.1 Proof of Proposition 3

*Proof.* Let

$$f(\boldsymbol{\mu}, u, \boldsymbol{\eta}) = -c(k,\rho,|\mathcal{A}|)\left(\sum_{a\in\mathcal{A}}\mu_a\left[w_a^{k^*}\right]\right)^{1/k^*} + \sum_{a\in\mathcal{A}}\eta_a,$$

where $w_a = (\eta_a - d_a u)_+$. By definition, we have

$$\begin{aligned}
&P\left(\left|\hat{\mathbf{T}}^*(v)(s) - \mathcal{T}^*(v)(s)\right| > t\right) \\
&\leq P\left(\sup_\pi\left|\hat{\mathbf{T}}^\pi(v)(s) - \mathcal{T}^\pi(v)(s)\right| > t\right) \\
&\leq P\left(\sup_{d\in\Delta(|\mathcal{A}|)}\gamma\left|\sup_{\boldsymbol{\eta}\in\mathbb{R}^{|\mathcal{A}|}}f(\boldsymbol{\mu}_n, R(s,\cdot,\cdot)+\gamma v, \boldsymbol{\eta}) - \sup_{\boldsymbol{\eta}\in\mathbb{R}^{|\mathcal{A}|}}f(\boldsymbol{\mu}, R(s,\cdot,\cdot)+\gamma v, \boldsymbol{\eta})\right| > t\right).
\end{aligned}$$

We analyze the sensitivity of the mapping $\boldsymbol{\mu} \to f(\boldsymbol{\mu}, u, \lambda)$. To control the difference between the empirical and the population objective, we establish the following lemma. The proof is deferred to Appendix D.2.

**Lemma 8.** *For any fixed $u$ and $\pi$,*

$$\left|\sup_{\boldsymbol{\eta}\in\mathbb{R}^{|\mathcal{A}|}}f(\boldsymbol{\mu}_n, u, \boldsymbol{\eta}) - \sup_{\boldsymbol{\eta}\in\mathbb{R}^{|\mathcal{A}|}}f(\boldsymbol{\mu}, u, \boldsymbol{\eta})\right| \leq c\|u\|_{\infty,\boldsymbol{\mu}}\left\|\frac{dm_n}{d\mu_n(t)}\right\|_{\infty,\boldsymbol{\mu}}$$

*where $c = 2^{1/(k-1)}k^*$.*

We decomposed the probability using the event $\Omega_{n,\delta_0}(\boldsymbol{\mu})$ where the empirical estimates are close to the population measures. Let $c = 2^{1/(k-1)}k^*$, and by using lemma 8, we obtain

$$P\left(\sup_{d\in\Delta(|\mathcal{A}|)}\gamma\left|\sup_{\boldsymbol{\eta}\in\mathbb{R}^{|\mathcal{A}|}}f(\boldsymbol{\mu}_n,u,\boldsymbol{\eta}) - \sup_{\boldsymbol{\eta}\in\mathbb{R}^{|\mathcal{A}|}}f(\boldsymbol{\mu},u,\boldsymbol{\eta})\right| > t\right)$$

$$\leq P(\Omega_{n,\delta_0}(\boldsymbol{\mu})^c) + P\left(c\,\|u\|_\infty\left\|\frac{dm_n}{d\mu_n(\tau)}\right\|_{\infty,\boldsymbol{\mu}} > t, \Omega_{n,\delta_0}(\boldsymbol{\mu})\right)$$

Again using Lemma 7, we have

$$\leq P\left(\sup_{a\in\mathcal{A},s'\in\mathcal{S}}\left|\frac{\mu_{a,n}(s')-\mu_a(s')}{\mu_a(s')}\right| > \delta_0\right) + P\left(c\|u\|_{\infty,\boldsymbol{\mu}}\sup_{a\in\mathcal{A},s'\in\mathcal{S}}\left|\frac{\mu_{a,n}(s')-\mu_a(s')}{(1-\delta_0)\mu_a(s')}\right| > t\right).$$

By Chernoff Bound and Bernstein Inequality, we obtain

$$\leq P\left(\sup_{a\in\mathcal{S},s'\in\mathcal{S}}\left|\frac{1}{n}\sum_{i=1}^n\mathbb{1}(S_i=s')-\mu_a(s')\right| > \delta_0\mu_a(s')\right)$$

$$+ P\left(\frac{c}{1-\delta_0}\|u\|_\infty\sup_{a\in\mathcal{S},s'\in\mathcal{S}}\frac{1}{\mu_a(s')}\left|\frac{1}{n}\sum_{i=1}^n\mathbb{1}(S_i=s')-\mu_a(s')\right| > t\right)$$

$$\leq 2\sum_{a\in\mathcal{A}}\sum_{s'\in\mathcal{S}}\left(\exp\left(-\frac{\delta_0^2 n\mu_a(s')}{3}\right) + \exp\left(-\frac{t^2}{2}\left(\frac{c^2\|u\|_\infty^2}{(1-\delta_0)^2 n\mu_a(s')} + \frac{c\|u\|_\infty t}{3(1-\delta_0)n\mu_a(s')}\right)^{-1}\right)\right),$$

Since $\mu_a(s') \geq \mathfrak{p}_\wedge$, and both exponential term above is monotonically decreasing over $\mu_a(s')$, we have

$$\leq 2|\mathcal{A}||\mathcal{S}|\left(\exp\left(-\frac{\delta_0^2 n\mathfrak{p}_\wedge}{3}\right) + \exp\left(-\frac{t^2}{2}\left(\frac{c^2\|u\|_\infty^2}{(1-\delta_0)^2 n\mathfrak{p}_\wedge} + \frac{c\|u\|_\infty t}{3(1-\delta_0)n\mathfrak{p}_\wedge}\right)^{-1}\right)\right)$$

Choose $\delta_0 = \frac{1}{2}$, by union bound, we obtain

$$P\left(\left\|\hat{\mathbf{T}}^*(v) - \mathcal{T}^*(v)\right\|_\infty > t\right)$$

$$\leq P\left(\sup_{s\in\mathcal{S}}\gamma\sup_{d\in\Delta(\mathcal{A})}\left|\sup_{\boldsymbol{\eta}\in\mathbb{R}^{|\mathcal{A}|}}f(P_{s,n},R(s,\cdot,\cdot)+\gamma v,\boldsymbol{\eta}) - \sup_{\boldsymbol{\eta}\in\mathbb{R}^{|\mathcal{A}|}}f(P_s,R(s,\cdot,\cdot)+\gamma v,\boldsymbol{\mu})\right| > t\right)$$

$$\leq 2|\mathcal{S}|^2|\mathcal{A}|\exp\left(-\frac{n\mathfrak{p}_\wedge}{12}\right) + 2|\mathcal{S}|^2|\mathcal{A}|\exp\left(-\frac{t^2}{2\gamma^2}\left(\frac{4c^2\|R(s,\cdot,\cdot)+\gamma v\|_\infty^2}{n\mathfrak{p}_\wedge} + \frac{2c\|R(s,\cdot,\cdot)+\gamma v\|_\infty t}{3\gamma n\mathfrak{p}_\wedge}\right)^{-1}\right).$$

Set each term to be less than $\eta/2$, by union bound, we need

$$n \geq \frac{12}{\mathfrak{p}_\wedge}\log\left(4|\mathcal{S}|^2|\mathcal{A}|/\eta\right) \tag{D.1}$$

$$t \geq \frac{4c\|R+\gamma v\|_\infty}{3n\mathfrak{p}_\wedge}\log\left(4|\mathcal{S}|^2|\mathcal{A}|/\eta\right) + \frac{2c\|R+\gamma v\|_\infty}{\sqrt{n\mathfrak{p}_\wedge}}\sqrt{2\log\left(4|\mathcal{S}|^2|\mathcal{A}|/\eta\right)}. \tag{D.2}$$

Under (D.1), we have

$$\frac{\log(4|\mathcal{S}|^2|\mathcal{A}|/\eta)}{n\mathfrak{p}_\wedge} \leq \sqrt{\frac{\log(4|\mathcal{S}|^2|\mathcal{A}|/\eta)}{n\mathfrak{p}_\wedge}}.$$

By substituting this bound into (D.2), we obtain

$$\frac{4c\|R+\gamma v\|_\infty}{3n\mathfrak{p}_\wedge}\log\left(4|\mathcal{S}|^2|\mathcal{A}|/\eta\right) + \frac{2c\|R+\gamma v\|_\infty}{\sqrt{n\mathfrak{p}_\wedge}}\sqrt{2\log\left(4|\mathcal{S}|^2|\mathcal{A}|/\eta\right)}$$

$$\leq \left(\frac{4c}{3} + 2\sqrt{2}c\right)\frac{\|R+\gamma v\|_\infty}{\sqrt{n\mathfrak{p}_\wedge}}\sqrt{\log\left(4|\mathcal{S}|^2|\mathcal{A}|/\eta\right)}$$

$$\leq \frac{3\cdot 2^{k^*}k^*\|R+\gamma v\|_\infty}{\sqrt{n\mathfrak{p}_\wedge}}\sqrt{\log\left(4|\mathcal{S}|^2|\mathcal{A}|/\eta\right)}.$$

Therefore, when $n$ satisfies (D.1) and $t$ satisfies

$$t \geq \frac{3 \cdot 2^{k^*} k^* \|R + \gamma v\|_\infty}{\sqrt{n \mathfrak{p}_\wedge}} \sqrt{\log \left(4|\mathcal{S}|^2|\mathcal{A}|/\eta\right)},$$

we have

$$P\left(\left\|\hat{\mathbf{T}}^*(v) - \mathcal{T}^*(v)\right\|_\infty > t\right) \leq \eta,$$

which implies the statement of the proposition. $\qquad\square$

## D.2 Proof of Lemma 8

*Proof.* We partition $\mathbb{R}^{|\mathcal{A}|}$ into three subsets, denote as

$$X_1 = \left\{ \boldsymbol{\eta} \middle| \eta_a \leq d_a \underset{\mu_a}{\operatorname{essinf}} u \text{ for all } a \in \mathcal{A} \right\},$$

$$X_2 = \left\{ \boldsymbol{\eta} \middle| \eta_a > d_a \underset{\mu_a}{\operatorname{essinf}} u \text{ for all } a \in \mathcal{A} \right\},$$

$$X_3 = \mathbb{R}^{|\mathcal{A}|} \backslash \{X_1 \cup X_2\}.$$

Next we prove that $X_3 = \varnothing$. If $\boldsymbol{\eta}$ is an optimal solution, then it satisfies the conditions described in the following lemma.

**Lemma 9.** *Let $\boldsymbol{\eta}^*(\boldsymbol{\mu})$ denote the optimal $\boldsymbol{\eta}$ under measure $\boldsymbol{\mu}$, then we have*

$$\left(\sum_{a \in \mathcal{A}} \mu_a \left[w_a^{k^*}\right]\right)^{1/k} = c(k, \rho, |\mathcal{A}|)\mu_i \left[w_i^{1/(k-1)}\right] \quad \text{for all } i \in \mathcal{A} \tag{D.3}$$

*and when $\boldsymbol{\eta} \in X_2$, we have*

$$f(\boldsymbol{\mu}, u, \boldsymbol{\eta}^*) = -\frac{\sum_{a \in \mathcal{A}} \mu_a \left[w_a^{k^*}\right]}{\mu_i \left[w_i^{1/(k-1)}\right]} + \sum_{a \in \mathcal{A}} \eta_a^* \quad \text{for any } i \in \mathcal{A}.$$

The proof is deferred to Appendix D.3. Suppose $\boldsymbol{\eta} \in X_3$. Then, there exists some $a' \in \mathcal{A}$ such that $\eta_{a'} \leq d_{a'} \operatorname{essinf}_{\mu_{a'}} u$, implying that $\mu_{a'} \left[w_{a'}^{1/(k-1)}\right] = 0$. According to (D.3), this leads to $\mu_a \left[w_a^{k^*}\right] = 0$ for all $a \in \mathcal{A}$, which means $\boldsymbol{\eta} \in X_1$, contradicting the initial assumption. Hence, $X_3 = \varnothing$.

When $\boldsymbol{\eta} \in X_1$, we have

$$\left| \sup_{\boldsymbol{\eta} \in X_1} f(\boldsymbol{\mu}_n, u, \boldsymbol{\eta}) - \sup_{\boldsymbol{\eta} \in X_1} f(\boldsymbol{\mu}, u, \boldsymbol{\eta}) \right| \leq \sup_{\boldsymbol{\eta} \in X_1} |f(\boldsymbol{\mu}_n, u, \boldsymbol{\eta}) - f(\boldsymbol{\mu}, u, \boldsymbol{\eta})|$$

$$= \left| \left(-0 + \sum_{a \in \mathcal{A}} \eta_a\right) - \left(-0 + \sum_{a \in \mathcal{A}} \eta_a\right) \right| = 0$$

Otherwise, $\boldsymbol{\eta} \in X_2$, for any fixed $\boldsymbol{\mu}, \boldsymbol{\mu}_n$ and $u$, let

$$g(\boldsymbol{\eta}, t) = f(\boldsymbol{\mu}_n(t), u, \boldsymbol{\eta}(\boldsymbol{\mu}_n(t))),$$
$$V(t) = \sup_{\boldsymbol{\eta} \in X_2} g(\boldsymbol{\eta}, t).$$

Before proceeding, we introduce the following version of the envelope theorem, which ensures the differentiability of $V(t)$ and provides an explicit formula for its derivative. This result allows us to apply the mean value theorem in the subsequent analysis.

**Lemma 10** (Envelope theorem, [11], Corollary 3)**.** *Denote $V$ as*

$$V(t) = \sup_{\mathbf{x} \in X} f(\mathbf{x}, t).$$

*Suppose that $X$ is a convex set in a linear space and $f : X \times [0, 1] \to \mathbb{R}$ is a concave function. Also suppose that $t_0 \in (0, 1)$, and that there is some $\mathbf{x}^* \in X^*(t_0)$ such that $d_t f(\mathbf{x}^*, t_0)$ exists. Then $V$ is differentiable at $t_0$ and $d_t V(t_0) = \partial_t f(\mathbf{x}^*, t_0)$*

We examine the convexity of $X$ and the concavity of $g$. $X_2$ is a convex set since it is defined by linear inequalities for each coordinate. For $g$, since $f$ serves as the dual objective function and is therefore concave, and concavity is preserved under affine mappings. So given that $\boldsymbol{\mu}_n(t)$ is a linear function of $t$, $g$ inherits the concavity. Therefore according to Lemma 10, $V(t)$ is differentiable. By mean value theorem, there exists $\tau \in (0, 1)$, for which the following equation holds:

$$\left| \sup_{\boldsymbol{\eta} \in X_2} f(\boldsymbol{\mu}_n, u, \boldsymbol{\eta}) - \sup_{\boldsymbol{\eta} \in X_2} f(\boldsymbol{\mu}, u, \boldsymbol{\eta}) \right| = \left| \sup_{\boldsymbol{\eta} \in X_2} g(\boldsymbol{\eta}, 0) - \sup_{\boldsymbol{\eta} \in X_2} g(\boldsymbol{\eta}, 1) \right|$$

$$= \left| \frac{d}{dt} V(t) \Big|_{t=\tau} \right|,$$

and by envelope theorem, we have

$$\frac{d}{dt} V(t) \Big|_{t=\tau} = \frac{\partial}{\partial t} g(\boldsymbol{\eta}^*, t) \Big|_{t=\tau}.$$

Recall that

$$g(\boldsymbol{\eta}, t) = -c(k, \rho, |\mathcal{A}|) \left( \sum_{a \in \mathcal{A}} \mu_{a,n}(t) \left[ w_a^{k^*} \right] \right)^{1/k^*} + \sum_{a \in \mathcal{A}} \eta_a,$$

by using (D.3), we obtain

$$\frac{\partial}{\partial t} g(\boldsymbol{\eta}^*, t) = -\frac{c(k, \rho, |\mathcal{A}|)}{\left( \sum_{a \in \mathcal{A}} \mu_{a,n}(t) \left[ w_a^{k^*} \right] \right)^{1/k}} \sum_{a \in \mathcal{A}} m_{a,n} \left[ w_a^{k^*} \right]$$

$$= -\frac{\sum_{a \in \mathcal{A}} m_{a,n} \left[ w_a^{k^*} \right]}{\mu_{i,n}(t) \left[ w_i^{1/(k-1)} \right]}.$$

Therefore,

$$\left| \frac{d}{dt} V(t) \Big|_{t=\tau} \right| = \left| \frac{\partial}{\partial t} g(\boldsymbol{\eta}^*(t), t) \Big|_{t=\tau} \right|$$

$$= \left| \frac{\sum_{a \in \mathcal{A}} m_{a,n} \left[ w_a^{k^*} \right]}{\mu_{i,n}(\tau) \left[ w_i^{1/(k-1)} \right]} \right|.$$

Since the equation above holds for any $i \in \mathcal{A}$, chose $i = a$ for each $a \in \mathcal{A}$, so we can rewrite the equation above as

$$\left| \sup_{\boldsymbol{\eta} \in \mathbb{R}^{|\mathcal{A}|}} f(\boldsymbol{\mu}_n, u, \boldsymbol{\eta}) - \sup_{\boldsymbol{\eta} \in \mathbb{R}^{|\mathcal{A}|}} f(\boldsymbol{\mu}, u, \boldsymbol{\eta}) \right| = \left| \sum_{a \in \mathcal{A}} \frac{m_{a,n} \left[ w_a^{k^*} \right]}{\mu_{a,n}(\tau) \left[ w_a^{1/(k-1)} \right]} \right|$$

$$\leq \sum_{a \in \mathcal{A}} \left| \frac{m_{a,n} \left[ w_a^{k^*} \right]}{\mu_{a,n}(\tau) \left[ w_a^{1/(k-1)} \right]} \right|.$$

For each term in the summation, we analyze $\eta_a \geq 2d_a \|u\|_{L^\infty(\mu_a)}$ and $d_a \operatorname{essinf}_{\mu_a} u \leq \eta_a < 2d_a \|u\|_{L^\infty(\mu_a)}$ separately. For $\eta_a \geq 2d_a \|u\|_{L^\infty(\mu_a)}$, by mean value theorem, there exists $\xi \in (\eta_a - d_a u, \eta_a)$ satisfies

$$\left| \frac{m_{a,n} \left[ w_a^{k^*} \right]}{\mu_{a,n}(\tau) \left[ w_a^{1/(k-1)} \right]} \right| = \left| \frac{m_{a,n} \left[ (\eta_a - d_a u)_+^{k^*} - (\eta_a)_+^{k^*} \right]}{\mu_{a,n}(\tau) \left[ (\eta_a - d_a u)_+^{1/(k-1)} \right]} \right|$$

$$= \left| \frac{m_{a,n} \left[ d_a u k^* (\xi)_+^{1/(k-1)} \right]}{\mu_{a,n}(\tau) \left[ (\eta_a - d_a u)_+^{1/(k-1)} \right]} \right|.$$

609    Since $\eta_a > 2d_a\|u\|_{L^\infty(\mu_a)}, \xi < \eta_a \le 2(\eta_a - d_a u)$, then we have

$$
\left| \frac{m_{a,n}\left[d_a u k^*(\xi)_+^{1/(k-1)}\right]}{\mu_{a,n}(\tau)\left[(\eta_a - d_a u)_+^{1/(k-1)}\right]} \right|
$$

$$
\le \left\| \frac{d_a u k^*(\xi)_+^{1/(k-1)}}{(\eta_a - d_a u)_+^{1/(k-1)}} \right\|_{L^\infty(\mu_a)} \left\| \frac{dm_n}{d\mu_n(\tau)} \right\|_{L^\infty(\mu_a)}
$$

$$
\le \left\| \frac{d_a u k^* 2^{1/(k-1)}(\eta_a - d_a u)_+^{1/(k-1)}}{(\eta_a - d_a u)_+^{1/(k-1)}} \right\|_{L^\infty(\mu_a)} \left\| \frac{dm_n}{d\mu_n(\tau)} \right\|_{L^\infty(\mu_a)}
$$

$$
= 2^{1/(k-1)} k^* d_a \|u\|_{L^\infty(\mu_a)} \left\| \frac{dm_n}{d\mu_n(\tau)} \right\|_{L^\infty(\mu_a)}.
$$

610    For $d_a \operatorname{essinf}_{\mu_a} u \le \eta_a < 2d_a\|u\|_{L^\infty(\mu_a)}, (\eta_a - d_a u)_+$ is bounded, then we have

$$
\left| \frac{m_{a,n}\left[w_a^{k^*}\right]}{\mu_{a,n}(\tau)\left[w_a^{1/(k-1)}\right]} \right| = \left| \frac{m_{a,n}\left[(\eta_a - d_a u)_+^{k^*}\right]}{\mu_{a,n}(\tau)\left[(\eta_a - d_a u)_+^{1/(k-1)}\right]} \right|
$$

$$
= \left\| \frac{(\eta_a - d_a u)_+ (\eta_a - d_a u)_+^{1/(k-1)}}{(\eta_a - d_a u)_+^{1/(k-1)}} \right\|_{L^\infty(\mu_a)} \left\| \frac{dm_n}{d\mu_n(\tau)} \right\|_{L^\infty(\mu_a)}
$$

$$
\le 2d_a \|u\|_{L^\infty(\mu_a)} \left\| \frac{dm_n}{d\mu_n(\tau)} \right\|_{L^\infty(\mu_a)}.
$$

611    To sum up, we have

$$
\sum_{a\in\mathcal{A}} \left| \frac{m_{a,n}\left[w_a^{k^*}\right]}{\mu_{a,n}(\tau)\left[w_a^{1/(k-1)}\right]} \right|
$$

$$
\le \sum_{a\in\mathcal{A}} \max\left\{ \sup_{\eta_a \ge 2d_a\|u\|_{L^\infty(\mu_a)}} \left| \frac{m_{a,n}\left[w_a^{k^*}\right]}{\mu_{a,n}(\tau)\left[w_a^{1/(k-1)}\right]} \right|, \sup_{\substack{d_a \operatorname{essinf}_{\mu_a} u \le \eta_a \\ \le 2d_a\|u\|_{L^\infty(\mu_a)}}} \left| \frac{m_{a,n}\left[w_a^{k^*}\right]}{\mu_{a,n}(\tau)\left[w_a^{1/(k-1)}\right]} \right| \right\}
$$

$$
= \sum_{a\in\mathcal{A}} \max\left\{ 2^{1/(k-1)}k^*, 2 \right\} d_a \|u\|_{L^\infty(\mu_a)} \left\| \frac{dm_n}{d\mu_n(\tau)} \right\|_{L^\infty(\mu_a)}
$$

$$
= 2^{1/(k-1)} k^* \|u\|_\infty \left\| \frac{dm_n}{d\mu_n(\tau)} \right\|_{\infty,\boldsymbol{\mu}}.
$$

612    Overall, let $c = 2^{1/(k-1)}k^*$, when $\boldsymbol{\mu} \in X_1$,

$$
\left| \sup_{\boldsymbol{\eta}\in\mathbb{R}^{|\mathcal{A}|}} f(\boldsymbol{\mu}_n, u, \boldsymbol{\eta}) - \sup_{\boldsymbol{\eta}\in\mathbb{R}^{|\mathcal{A}|}} f(\boldsymbol{\mu}, u, \boldsymbol{\eta}) \right| = 0 \le c\|u\|_\infty \left\| \frac{dm_n}{d\mu_n(\tau)} \right\|_{\infty,\boldsymbol{\mu}},
$$

613    when $\boldsymbol{\mu} \in X_2$,

$$
\left| \sup_{\boldsymbol{\eta}\in\mathbb{R}^{|\mathcal{A}|}} f(\boldsymbol{\mu}_n, u, \boldsymbol{\eta}) - \sup_{\boldsymbol{\eta}\in\mathbb{R}^{|\mathcal{A}|}} f(\boldsymbol{\mu}, u, \boldsymbol{\eta}) \right| \le \sum_{a\in\mathcal{A}} \left| \frac{m_{a,n}\left[w_a^{k^*}\right]}{\mu_{a,n}(\tau)\left[w_a^{1/(k-1)}\right]} \right| \le c\|u\|_\infty \left\| \frac{dm_n}{d\mu_n(\tau)} \right\|_{\infty,\boldsymbol{\mu}}.
$$

614    $\qquad\qquad\qquad\qquad\qquad\qquad\qquad\qquad\qquad\qquad\qquad\qquad\qquad\qquad\qquad\qquad\qquad$ $\square$

### D.3 Proof of Lemma 9

Since $f(\boldsymbol{\mu}, u, \boldsymbol{\eta}^*)$ is the objective function of the dual problem and $\boldsymbol{\eta}$ is the dual variable, $f$ is convex with respect to $\boldsymbol{\eta}$. To optimize $f$ over $\boldsymbol{\eta}$, we set its derivative with respect to $\eta_i$ to zero, which yields

$$\frac{\partial}{\partial \eta_i^*} f(\boldsymbol{\mu}, u, \boldsymbol{\eta}^*) = 1 - c(k, \rho, |\mathcal{A}|) \left( \sum_{a \in \mathcal{A}} \mu_a \left[ w_a^{k^*} \right] \right)^{-1/k} \mu_i \left[ w_i^{1/(k-1)} \right] = 0,$$

which means

$$\left( \sum_{a \in \mathcal{A}} \mu_a \left[ w_a^{k^*} \right] \right)^{1/k} = c(k, \rho, |\mathcal{A}|) \mu_i \left[ w_i^{1/(k-1)} \right] \quad \text{for all } i \in \mathcal{A},$$

which is (D.3). When $\boldsymbol{\eta} \in X_2$, $\mu_i[w_i^{1/(k-1)}]$ is positive, plug in $\eta_a^*$, we obtain

$$f(\boldsymbol{\mu}, u, \boldsymbol{\eta}) = -c(k, \rho, |\mathcal{A}|) \left( \sum_{a \in \mathcal{A}} \mu_a \left[ w_a^{k^*} \right] \right)^{1/k^*} + \sum_{a \in \mathcal{A}} \eta_a$$

$$= -\frac{\sum_{a \in \mathcal{A}} \mu_a \left[ w_a^{k^*} \right]}{\mu_i \left[ w_i^{1/(k-1)} \right]} + \sum_{a \in \mathcal{A}} \eta_a^*.$$

### D.4 Proof of Theorem 2

*Proof.* Substituting $\|R + \gamma v\|_\infty \leq 1/(1 - \gamma)$ into the bound from Proposition 2 and applying Proposition 1, we obtain the stated result.

$$\|\hat{v} - v^*\|_\infty \leq \frac{1}{1 - \gamma} \left\| \hat{\mathbf{T}}^*(v^*) - \mathcal{T}^*(v^*) \right\|_\infty$$

$$\leq \frac{3 \cdot 2^{k^*} k^* \|R + \gamma v\|_\infty}{(1 - \gamma) \sqrt{n \mathfrak{p}_\wedge}} \sqrt{\log \left( 4|\mathcal{S}|^2 |\mathcal{A}| / \eta \right)}$$

$$\leq \frac{3 \cdot 2^{k^*} k^*}{(1 - \gamma)^2 \sqrt{n \mathfrak{p}_\wedge}} \sqrt{\log \left( 4|\mathcal{S}|^2 |\mathcal{A}| / \eta \right)}$$

with probability $1 - \eta$. $\qquad \square$

