# OpenReview forum: "Near-Optimal Sample Complexities of  Divergence-based  S-rectangular Distributionally Robust Reinforcement Learning"
_NeurIPS.cc/2025/Conference — Submitted to NeurIPS 2025_

### Official Review · Reviewer_bgCq · 2025-06-07

**Clarity:** 3
**Significance:** 3
**Originality:** 3
**Rating:** 3
**Confidence:** 3

**Summary:**

This work obtains the first near-optimal sample complexity bound $\widetilde{O}[|S||A|(1-\gamma)^{-4}\epsilon^{-2}]$ of empirical value iteration algorithm on distributionally robust reinforcement learning (DR-RL) problem with $f$-divergence based $s$-rectangular uncertainty set.

**Questions:**

(1) What's the theoretical complexity lower bound? You could write it down and cite the corresponding work, in the introduction and other proper places. The conclusion section mentions the complexity gap as a limitation. Does it mean dependence on $1-\gamma$ is not optimal? You might make it more explicit.

(2) Could you cite the following works of value iteration on robust MDP? Also, [1] seems to have better dependence on $\epsilon$. How to explain?

[1] Grand-Clément, J. and Kroer, C. (2021). Scalable first-order methods for robust mdps. In Proceedings of the AAAI Conference on Artificial Intelligence, volume 35, pages 12086–12094.

[2] Kumar, N., Levy, K., Wang, K., and Mannor, S. (2023b). An efficient solution to s-rectangular robust markov decision processes. ArXiv:2301.13642.

(3) Are Lemmas 1 and 2 proposed by you or existing literature? If latter, please cite.

(4) To reproduce Section 5.2 (the second experiment), what does "estimation error" mean? (might be defined by equations) What is the discount factor $\gamma$?

Minor issues:

(5) $A_t$ instead of $a_t$ in Examples 1.

(6) $\|R+\gamma v\| _ {\infty}$ in Proposition 2 could be defined, especially considering that $R(s,a)$ and $v(s)$ take different input variables.

(7) In Section 5, does $n_0=n$?

(8) Figure 4 caption: "number of actions".

**Ethical Concerns:**

["NO or VERY MINOR ethics concerns only"]

**Final Justification:**

I just read the interaction between the authors and the other reviewers.

I also agree with Reviewer Qk39 that this work is not significant. This work simply changes transition kernel from P into its empirical estimation P_hat, which has no significant technical challenge and novelty. Also this work only focuses on statistical

So I reduce my rating to 3.

**Limitations:**

Yes, the authors have written their two limitations in the conclusion section: (1) "The reliance on access to a generative model and the presence of a gap between our upper bound and the minimax lower bound established in Yang [29]"; (2) "As part of future work, we aim to develop provable theoretical guarantees for other settings, including model-free algorithms and offline reinforcement learning."

**Paper Formatting Concerns:**

I just read the interaction between the authors and the other reviewers.

I also agree with Reviewer Qk39 that this work is not significant. This work simply changes transition kernel P into P_hat, which has no significant technical challenge and novelty.

So I reduce my rating to 3.

Reviewer bgCq

**Quality:**

3

**Strengths And Weaknesses:**

Strengths: This is a complete and clear work. The topic of distributionally robust reinforcement learning is popular with many applications. It achieves the first theoretical sample complexity result with optimal dependence on $\epsilon,|S|,|A|$, and the complexity result is verified by simulations. The result looks reasonable to me based on my knowledge of value iteration and sample-based transition kernel estimation. The proof has improved the existing techniques from [25]. This work may be cited in the future in either of two ways: (1) Apply this algorithm to small state-action spaces due to efficient sample complexity obtained by this work. (2) Improve this work by future works for more practical and tricky settings such as large state-action spaces, lack of generative model, model-free setting, offline reinforcement learning, etc.

Weaknesses: The second experiment in Section 5.2 needs more clarification to be reproducible (See my questions below). It also would be better to compare with other existing sample-based value iteration algorithms in the experiments. A few more citations are needed (See my questions below). There are some limitations as listed in the conclusion section.

---

> ### Author Rebuttal · Authors · 2025-07-31
>
> We thank the reviewer for their comments and valuable suggestions. We address their questions below.
>
> ### Q1. What's the theoretical complexity lower bound?
>
> The theoretical complexity lower bound under the $\chi^2$-divergence uncertainty set is given in [1], which is
>
> $$
> \tilde{\Omega}\Big(\frac{|\mathcal{S}||\mathcal{A}|}{\varepsilon^2(1-\gamma)^2}\min\Big\\{\frac1{1-\gamma},\frac1\rho\Big\\}\Big).
> $$
> We currently have not included a direct table comparing our results with this lower bound, but will mention and compare it in the discussion section of the revised version.
>
> [1] Yang, W., Zhang, L., & Zhang, Z. (2022). Toward theoretical understandings of robust markov decision processes: Sample complexity and asymptotics. *The Annals of Statistics*, *50*(6), 3223-3248.
>
>
>
> ### Q2.Could you cite the following works of value iteration on robust MDP? Also, [1] seems to have better dependence on $\epsilon$. How to explain?
>
> Thank you for your question and literature suggestion. We will cite the related works [1,2] on solving robust MDPs.
> Regarding the $\epsilon$-dependence, [1] and [2] focus on an optimization setting where the underlying robust MDP instance is known to the optimizer. In contrast, our setting assumes an unknown robust MDP instance where we need to learn an optimal robust policy from data. Therefore, [1,2] report the **iteration complexity**,  whereas our paper analyzes the **statistical complexity**. Since these two types of complexity measure different aspects of the problem, their dependence on $\epsilon$ is not directly comparable.
>
> [1] Grand-Clément, J. and Kroer, C. (2021). Scalable first-order methods for robust mdps. In Proceedings of the AAAI Conference on Artificial Intelligence, volume 35, pages 12086–12094.
>
> [2] Kumar, N., Levy, K., Wang, K., and Mannor, S. (2023b). An efficient solution to s-rectangular robust markov decision processes. ArXiv:2301.13642.
>
>
>
> ### Q3.Are Lemmas 1 and 2 proposed by you or existing literature? If latter, please cite.
>
> A similar result was presented in previous work [1] where $R$ is only a function of $S_t,A_t$. However, since we consider the case where $R(S_t, A_t, S_{t+1})$ can depend on the next state (which is important to many application settings, e.g., inventory problems), the results in [1] cannot be applied directly. So, we prove these lemmas in this paper.
>
> [1] Yang, W., Zhang, L., & Zhang, Z. (2022). Toward theoretical understandings of robust markov decision processes: Sample complexity and asymptotics. *The Annals of Statistics*, *50*(6), 3223-3248.
>
>
>
> ### Q4. To reproduce Section 5.2 (the second experiment), what does "estimation error" mean? (might be defined by equations) What is the discount factor $\gamma$?
>
> Thank you for pointing out this clarity issue. The “estimation error” is defined as
>
> $$
> \text{estimation error} = \left\Vert\max_{\pi} V_{\mathcal{P}}^{\pi} - \max_{\pi} V_{\hat{\mathcal{P}}}^{\pi} \right\Vert_{\infty},
> $$
>
> where $V_{\mathcal{P}}^{\pi}$ and $V_{\hat{\mathcal{P}}}^{\pi}$ denote the value functions under the true and estimated transition probability, respectively.
> We thank the reviewer for pointing out the omission of the discount factor. In our experiments, we set $\gamma = 0.9$.
> We have added these clarifications in the revision.
>
>
> ### Q6. $\Vert R+\gamma v\Vert_{\infty}$ in Proposition 2 could be defined, especially considering that $R(s,a)$  and $v(s)$ take different input variables.
>
> Thanks for pointing this out. Here, $\Vert R+\gamma v\Vert_{\infty}$ is defined as
> $$
> \Vert R+\gamma v\Vert_{\infty} = \sup_{s,a,s^\prime}|R(s,a,s^\prime)+\gamma v(s^\prime)|
> $$
>
> where $R(s, a, s')$ is the reward function and $v(s')$ is the value function at state $s'$.
> We will make it clear in the revised version.
>
> ### Q7. In Section 5, does $n_0=n$ ?
>
> No, $n_0$ and $n$ are not the same. Here, $n$ denotes the total number of samples, while $n_0$ is the number of samples collected for each $(s,a)$ pair. They are related by
> $$
> n = |\mathcal{S}| \cdot |\mathcal{A}| \cdot n_0.
> $$
>
> We have clarified this in the revision.
>
> ### Q5.Q8. $A_t$ insead of $a_t$ in Example 1.  Figure 4 caption: "number of actions".
> We thank the reviewer for carefully pointing out these typos. We will correct  them in the revision.

---

> ### Comment · Reviewer_bgCq · 2025-08-03
> **My questions are well solved**
>
> Thank the authors for your response.
> My questions are well solved.
> Since the theoretical result is obtained using empirical transition evaluation without significant challenge, I plan to keep my rating for now.

---

> ### Author Response · Authors · 2025-08-05
>
> Dear reviewer bgCq,
>
> Thank you for your feedback and thoughtful review.
>
> Best regards,
> The authors

---

### Official Review · Reviewer_Qk39 · 2025-06-27

**Clarity:** 3
**Significance:** 2
**Originality:** 1
**Rating:** 2
**Confidence:** 4

**Summary:**

The paper establishes sample complexity of value iteration for divergence based s-rectangular robust MDPs. It assumes the generative model of the MDP, and constructs empirical nominal kernels using the samples, then use this for Bellman updates.

**Questions:**

***Q1)  Para (Line 40-44) of the paper discusses the nature of the optimal policies for SA/S -rect. Robust MDPs. The should mention that the optimal policies for s-rect can be stochastic [1] and a exact nature (threshold power policy) was shown in [2] for L_p bounded s-rect MDP.***

***Q2) Literature review missing mention large bodies of relevant works such as s-rectangular works [2,4,8] ,  non -rectangular works  [3,5] that are even less convervative than s-rect,  KL-based SA-rect work [6], and [7] that has sample complexity for R-contamination uncertainty sets***

***Q3) Please compare your techniques and results with [9,10], it also claims $O(\epsilon^{-2})$ sample complexity for policy evaluation  diverence based robust MDPs. Can policy evaluation be trivially used to get optimal value funtion using policy iteration ?***

[1] Wolfram Wiesemann, Daniel Kuhn, Berç Rustem, (2012) Robust Markov Decision Processes. Mathematics of Operations Research 38(1):153-183.
https://doi.org/10.1287/moor.1120.0566

[2]@inproceedings{
kumar2024efficient,
title={Efficient Value Iteration for s-rectangular Robust Markov Decision Processes},
author={Navdeep Kumar and Kaixin Wang and Kfir Yehuda Levy and Shie Mannor},
booktitle={Forty-first International Conference on Machine Learning},
year={2024},
url={https://openreview.net/forum?id=J4LTDgwAZq}
}

[3] Gadot, U., Derman, E., Kumar, N., Elfatihi, M. M., Levy, K., & Mannor, S. (2024). Solving Non-rectangular Reward-Robust MDPs via Frequency Regularization. Proceedings of the AAAI Conference on Artificial Intelligence, 38(19), 21090-21098. https://doi.org/10.1609/aaai.v38i19.30101

[4]@inproceedings{10.5555/3666122.3668721,
author = {Kumar, Navdeep and Derman, Esther and Geist, Matthieu and Levy, Kfir and Mannor, Shie},
title = {Policy gradient for rectangular robust Markov decision processes},
year = {2023},
publisher = {Curran Associates Inc.},
address = {Red Hook, NY, USA},
booktitle = {Proceedings of the 37th International Conference on Neural Information Processing Systems},
articleno = {2599},
numpages = {25},
location = {New Orleans, LA, USA},
series = {NIPS '23}
}

[5]@misc{kumar2025dualformulationnonrectangularlp,
      title={Dual Formulation for Non-Rectangular Lp Robust Markov Decision Processes},
      author={Navdeep Kumar and Adarsh Gupta and Maxence Mohamed Elfatihi and Giorgia Ramponi and Kfir Yehuda Levy and Shie Mannor},
      year={2025},
      eprint={2502.09432},
      archivePrefix={arXiv},
      primaryClass={cs.AI},
      url={https://arxiv.org/abs/2502.09432},
}

[6] @inproceedings{
gadot2024bring,
title={Bring Your Own (Non-Robust) Algorithm to Solve Robust {MDP}s by Estimating The Worst Kernel},
author={Uri Gadot and Kaixin Wang and Navdeep Kumar and Kfir Yehuda Levy and Shie Mannor},
booktitle={Forty-first International Conference on Machine Learning},
year={2024},
url={https://openreview.net/forum?id=UqoG0YRfQx}
}

[7] @inproceedings{
wang2021online,
title={Online Robust Reinforcement Learning with Model Uncertainty},
author={Yue Wang and Shaofeng Zou},
booktitle={Advances in Neural Information Processing Systems},
editor={A. Beygelzimer and Y. Dauphin and P. Liang and J. Wortman Vaughan},
year={2021},
url={https://openreview.net/forum?id=IhiU6AJYpDs}
}

[8]@misc{derman2021twiceregularizedmdpsequivalence,
      title={Twice regularized MDPs and the equivalence between robustness and regularization},
      author={Esther Derman and Matthieu Geist and Shie Mannor},
      year={2021},
      eprint={2110.06267},
      archivePrefix={arXiv},
      primaryClass={cs.LG},
      url={https://arxiv.org/abs/2110.06267},
}

[9]@misc{xu2025finitesampleanalysispolicyevaluation,
      title={Finite-Sample Analysis of Policy Evaluation for Robust Average Reward Reinforcement Learning},
      author={Yang Xu and Washim Uddin Mondal and Vaneet Aggarwal},
      year={2025},
      eprint={2502.16816},
      archivePrefix={arXiv},
      primaryClass={stat.ML},
      url={https://arxiv.org/abs/2502.16816},
}

[10]@misc{xu2025efficientqlearningactorcriticmethods,
      title={Efficient $Q$-Learning and Actor-Critic Methods for Robust Average Reward Reinforcement Learning},
      author={Yang Xu and Swetha Ganesh and Vaneet Aggarwal},
      year={2025},
      eprint={2506.07040},
      archivePrefix={arXiv},
      primaryClass={cs.LG},
      url={https://arxiv.org/abs/2506.07040},
}

**Ethical Concerns:**

["NO or VERY MINOR ethics concerns only"]

**Final Justification:**

In absence of proper comaparision of relevant literature, I am not able to judge the novelty very clearly. Additionally, there exist similar results for non-robust counterpart and I am not convinced that its extension to robust, is significant (technically or conceptually). Hence, I maintain my rating of 2.

**Limitations:**

OK

**Paper Formatting Concerns:**

All seems ok.

**Quality:**

2

**Strengths And Weaknesses:**

Strengths: Established $O(\epsilon^{-2})$ sample complexity of the divergence based robust MDPs.

Weakness: 1) The approach builds empirical kernel using samples which is not feasible for large states space MDPs. 2) Missing large body of relevant works, see questions.

---

> ### Author Rebuttal · Authors · 2025-07-31
>
> We thank the reviewer for their comments and valuable suggestions. We address their questions below.
>
> ### Q1. The approach builds empirical kernel using samples which is not feasible for large states space MDPs.
> We thank the reviewer for their feedback. However, we want to remark that model-based methods that uses an empirical kernel or empirical measures are important cases for not only understanding the statistical complexity but also practical implementation. This is because, in many high-risk real-world settings such as aerospace engineering, finance, or health care, we usually cannot rely on function approximation. Moreover, from a sample-efficiency perspective, empirical kernel-based methods usually enjoys better efficiency, hence are suitable for applications where the data is scarce [1]. On the other hand, we are considering a generative model setting, and it is usually a first step to study model-based methods, as they are typically more efficient. Moreover, the analysis of these model-based methods is an important building blocks for other application-specific designs in future research.
>
> [1] Chua, Kurtland, et al. "Deep reinforcement learning in a handful of trials using probabilistic dynamics models." Advances in neural information processing systems 31 (2018).
>
> ### Q2.Para (Line 40-44) do not mention the optimal policies for S-rect can be stochastic and has a threshold power policy nature.
>
> We have mentioned that the optimal policies for S-rectangular models can be stochastic in lines 60-66. We will add this fact around Example 1.
>
> ### Q3. Literature review missing mention large bodies of relevant works such as s-rectangular works [2,4,8] , non -rectangular works [3,5] that are even less convervative than s-rect, KL-based SA-rect work [6], and [7] that has sample complexity for R-contamination uncertainty sets
>
> We thank the reviewer for pointing out these important and relevant works [2, 3, 4, 5, 6, 7, 8]. These works primarily focus on optimization, while we mainly focus on the statistical aspects. We will make sure to cite and properly acknowledge them in the revised version.
>
> [9,10] focus on **average-reward SA-rectangular** robust RL, while we focus on **discounted-reward S-rectangular** robust RL. Furthermore, [10] is uploaded after Neurips submission deadline and the first version of  [9] has some mistakes. Therefore, there are not relevant to this paper.
>
> [2]@inproceedings{ kumar2024efficient, title={Efficient Value Iteration for s-rectangular Robust Markov Decision Processes}, author={Navdeep Kumar and Kaixin Wang and Kfir Yehuda Levy and Shie Mannor}, booktitle={Forty-first International Conference on Machine Learning}, year={2024}, url={https://openreview.net/forum?id=J4LTDgwAZq} }
>
> [3] Gadot, U., Derman, E., Kumar, N., Elfatihi, M. M., Levy, K., & Mannor, S. (2024). Solving Non-rectangular Reward-Robust MDPs via Frequency Regularization. Proceedings of the AAAI Conference on Artificial Intelligence, 38(19), 21090-21098. https://doi.org/10.1609/aaai.v38i19.30101
>
> [4]@inproceedings{10.5555/3666122.3668721, author = {Kumar, Navdeep and Derman, Esther and Geist, Matthieu and Levy, Kfir and Mannor, Shie}, title = {Policy gradient for rectangular robust Markov decision processes}, year = {2023}, publisher = {Curran Associates Inc.}, address = {Red Hook, NY, USA}, booktitle = {Proceedings of the 37th International Conference on Neural Information Processing Systems}, articleno = {2599}, numpages = {25}, location = {New Orleans, LA, USA}, series = {NIPS '23} }
>
> [5]@misc{kumar2025dualformulationnonrectangularlp, title={Dual Formulation for Non-Rectangular Lp Robust Markov Decision Processes}, author={Navdeep Kumar and Adarsh Gupta and Maxence Mohamed Elfatihi and Giorgia Ramponi and Kfir Yehuda Levy and Shie Mannor}, year={2025}, eprint={2502.09432}, archivePrefix={arXiv}, primaryClass={cs.AI}, url={https://arxiv.org/abs/2502.09432}, }
>
> [6] @inproceedings{ gadot2024bring, title={Bring Your Own (Non-Robust) Algorithm to Solve Robust {MDP}s by Estimating The Worst Kernel}, author={Uri Gadot and Kaixin Wang and Navdeep Kumar and Kfir Yehuda Levy and Shie Mannor}, booktitle={Forty-first International Conference on Machine Learning}, year={2024}, url={https://openreview.net/forum?id=UqoG0YRfQx} }
>
> [7] @inproceedings{ wang2021online, title={Online Robust Reinforcement Learning with Model Uncertainty}, author={Yue Wang and Shaofeng Zou}, booktitle={Advances in Neural Information Processing Systems}, editor={A. Beygelzimer and Y. Dauphin and P. Liang and J. Wortman Vaughan}, year={2021}, url={https://openreview.net/forum?id=IhiU6AJYpDs} }
>
> [8]@misc{derman2021twiceregularizedmdpsequivalence, title={Twice regularized MDPs and the equivalence between robustness and regularization}, author={Esther Derman and Matthieu Geist and Shie Mannor}, year={2021}, eprint={2110.06267}, archivePrefix={arXiv}, primaryClass={cs.LG}, url={https://arxiv.org/abs/2110.06267}, }
>
> [9]@misc{xu2025finitesampleanalysispolicyevaluation, title={Finite-Sample Analysis of Policy Evaluation for Robust Average Reward Reinforcement Learning}, author={Yang Xu and Washim Uddin Mondal and Vaneet Aggarwal}, year={2025}, eprint={2502.16816}, archivePrefix={arXiv}, primaryClass={stat.ML}, url={https://arxiv.org/abs/2502.16816}, }
>
> [10]@misc{xu2025efficientqlearningactorcriticmethods, title={Efficient
> -Learning and Actor-Critic Methods for Robust Average Reward Reinforcement Learning}, author={Yang Xu and Swetha Ganesh and Vaneet Aggarwal}, year={2025}, eprint={2506.07040}, archivePrefix={arXiv}, primaryClass={cs.LG}, url={https://arxiv.org/abs/2506.07040}, }

---

> > ### Comment · Reviewer_Qk39 · 2025-08-03
> >
> > Thanks for the response. I am still not convinced about the significance of the results. Additionally, I would like the comparision and challenges between this work and its counterpart non-robust MDPs.

---

> > > ### Author Response · Authors · 2025-08-04
> > > **Thanks for the questions**
> > >
> > > Thank you for your question. We are happy to provide further clarification below.
> > >
> > > **Significance of this paper:**
> > > We present the *first* near-optimal sample complexity result for divergence-based S-rectangular distributionally robust reinforcement learning (DR-RL). Our result achieves *linear* dependence on the size of the state-action space, which significantly improves upon the only existing result in this setting [1], where the sample complexity scales as \$\tilde{O}(|\mathcal{S}|^2|\mathcal{A}|^2)\$.
> > >
> > > **Comparison with and challenges beyond standard (non-robust) MDPs:**
> > > Our analysis departs from that of classical (non-robust) MDPs in several key ways, introducing unique technical challenges:
> > >
> > > 1. **Randomized policies:** In the robust setting, the optimal policy may be *randomized*. This invalidates the common analysis route via Q-functions, which typically applies to deterministic policies. Instead, we must analyze a much richer policy class that includes *all* randomized policies, greatly increasing the complexity of the space under consideration.
> > >
> > > 2. **Minimum support assumption under KL-divergence:**
> > >    To achieve \$\tilde{O}(\varepsilon^{-2})\$ sample complexity under KL-divergence uncertainty sets, one must assume a *minimum support probability* $\mathfrak{p}_\wedge$. Without this assumption, such guarantees are information-theoretically impossible. A similar phenomenon was observed in the static setting \[2], and incorporating this into the analysis further complicates the proof structure.
> > >
> > >
> > > [1] Yang, Wenhao, Liangyu Zhang, and Zhihua Zhang. "Toward theoretical understandings of robust Markov decision processes: Sample complexity and asymptotics." *Annals of Statistics*, 50(6), 2022.
> > > [2] Duchi, John C., and Hongseok Namkoong. "Learning models with uniform performance via distributionally robust optimization." *Annals of Statistics*, 49(3), 2021.

---

> > > > ### Comment · Reviewer_Qk39 · 2025-08-07
> > > >
> > > > Thanks for the response. In absence of proper comaparision of relevant literature, I am not able to judge the novelty very clearly.  Additionally, there exist similar results for non-robust counterpart and I am not convinced that its extension to robust,  is significant (technically or conceptually). Hence, I maintain my rating of 2.

---

> ### Author Response · Authors · 2025-08-05
>
> Dear Reviewer Qk39,
>
> We hope that we have addressed all of your concerns. As the author-reviewer discussion period concludes, please do not hesitate to reach out if further clarifications are needed. We would be delighted to provide additional explanations or adjustments.
>
> Thank you again for your invaluable feedback and contributions to this work.
>
> Best wishes,
> The authors

---

### Official Review · Reviewer_5h1f · 2025-06-30

**Clarity:** 1
**Significance:** 2
**Originality:** 2
**Rating:** 3
**Confidence:** 3

**Summary:**

This paper addresses distributionally robust RL in the S-rectangular setting. This paper establishes a statistical bound on the *estimation error* for the optimal robust value. However, it does not address the *sample complexity*.

**Questions:**

1. If I understand correctly, Theorems 1 and 2 are **not** the *sample complexity* discussed in [1, 2]. According to Proposition 1, $\widehat{v} = \widehat{T}^*(\widehat{v})$ should refer to the optimal value under the **empirical DR Bellman Operators (Def.3)**, which is not the optimal robust value function defined w.r.t the true uncertainty set (i.e., $V_{\mathcal{P}}^{\widehat{\pi}}$). If that is true, I believe it is inappropriate to claim your contributions as "the first sample complexity results for divergence-based S-rectangular models".
2. I think the reference in Line 332 should be [30], instead of [29].
3. Can you explain why the bounds in Theorems 1 and 2 do not involve the uncertainty radius $\rho$?


[1] Clavier, P., Shi, L., Le Pennec, E., Mazumdar, E., Wierman, A., & Geist, M. (2024). Near-Optimal Distributionally Robust Reinforcement Learning with General $L_p$ Norms. Advances in Neural Information Processing Systems.

[2] Yang, W., Zhang, L., & Zhang, Z. (2022). Toward theoretical understandings of robust Markov decision processes: Sample complexity and asymptotics. The Annals of Statistics.

**Ethical Concerns:**

["NO or VERY MINOR ethics concerns only"]

**Final Justification:**

After reading the authors' response, I have raised my score accordingly. However, I remain skeptical about the tightness of the derived bounds due to their independence of the uncertainty radius.

**Limitations:**

yes

**Quality:**

2

**Strengths And Weaknesses:**

* **Strengths:**
    1. The writing of this paper is clear,
    2. Numerical results are presented and align with the theoretical analysis.

* **Weaknesses:**
    1. The definitions are unclear (See Question 1 below).
    2. The involvement of the minimum support probability (Def. 5) for $f_k$-divergence uncertainty set is sub-optimal when compared to the previously known bound [Theorem 3.2(b), 1].

[1] Yang, W., Zhang, L., & Zhang, Z. (2022). Toward theoretical understandings of robust Markov decision processes: Sample complexity and asymptotics. The Annals of Statistics.

---

> ### Author Rebuttal · Authors · 2025-07-31
>
> We thank the reviewer for their comments and valuable suggestions.
> ### Q1. Theorems 1 and 2 are **not** the *sample complexity* discussed in [1, 2].
> We thank the reviewer for their input. We want to remark that the sample complexity of learning the optimal value function within an absolute error of $\epsilon$ is an important question by itself [4]. In particular, they are useful for the analysis of value-based procedures, as well as for certifying the performance of the robust optimal policy. With this said, we recognize that the sample complexity of obtaining an $\epsilon$ optimal policy is also important and is not hard to obtain from our analysis. So, we enrich our paper by proving the following theorem that aligns with the sample complexity discussion in [1,2]. In particular, we show the following theorem 3 in the revision. Recall that
> $$
> V_{\mathcal{P}}^\pi(s) = \inf_{P\in\mathcal{P}}E_{P}^\pi \left[\left.\sum_{t=0}^\infty \gamma^t R(S_t, A_t,S_{t+1})\right|{S_0 = s}\right]
> $$
> >**Theorem 3**
> > Under the KL-divergence uncertainty set, with probability $1-\eta$
> >$$\sup_{\pi\in\Pi}V_{\mathcal{P}}^{\pi}-V_{\mathcal{P}}^{\hat{\pi}^\*}\leq\frac{18}{\sqrt{n\mathfrak{p}}(1-\gamma)^2}\sqrt{\log{(4|\mathcal{S}|^2|\mathcal{A}|/\eta)}}.$$
> >Under $f_k$-divergence uncertainty set, with probability $1-\eta$
> >$$
> \sup_{\pi\in\Pi}V_{\mathcal{P}}^{\pi}-V_{\mathcal{P}}^{\hat{\pi}^\*}\leq\frac{6\cdot2^{k^\*}k^\*}{\sqrt{n\mathfrak{p}}(1-\gamma)^2}\sqrt{\log\left(4|S|^2|\mathcal{A}|/\eta\right)}.
> $$
>
> We remark that showing a tight rate for the convergence of the estimated value function in Theorems 1 and 2 is more challenging than in Theorem 3, due to the max operation over randomized policies. The proof of Theorem 3 is provided here.
>
> **Proof of Theorem 3**
> We first show that $\sup_{\pi\in\Pi}\Vert\hat{\mathbf{T}}^{\pi}(v)-\mathcal{T}^{\pi}(v)\Vert_{\infty}$ shares the same upper bound as $\Vert\hat{\mathbf{T}}^\*(v)-\mathcal{T}^\*(v)\Vert_{\infty}$ . Specifically, under KL-divergence uncertainty set, with probability $1-\eta$
> $$
> \sup_{\pi\in\Pi}\left\Vert\hat{\mathbf{T}}^{\pi}(v)-\mathcal{T}^{\pi}(v)\right\Vert_{\infty}\leq \frac{9\Vert R+\gamma v\Vert_\infty}{\sqrt{n\mathfrak{p}}}\sqrt{\log{(4|\mathcal{S}|^2|\mathcal{A}|/\eta)}},
> $$
> and under $f_k$-divergence uncertainty set, with probability $1-\eta$
> $$
> \sup_{\pi\in\Pi}\left\Vert\hat{\mathbf{T}}^{\pi}(v)-\mathcal{T}^{\pi}(v)\right\Vert_{\infty}\leq \frac{3\cdot2^{k^\*}k^\*\Vert R+\gamma v\Vert_{\infty}}{\sqrt{n\mathfrak{p}}}\sqrt{\log\left(4|S|^2|\mathcal{A}|/\eta\right)}.
> $$
> This is because, in the proofs of Proposition 2 and Proposition 3, we first relax the probability bound from $P(|\hat{\mathbf{T}}^\*(v)(s) - \mathcal{T}^\*(v)(s)| > t)$ to $P(\sup_{\pi}|\hat{\mathbf{T}}^\pi(v)(s) - \mathcal{T}^\pi(v)(s)| > t)$, and then take the supremum over states and actions in subsequent analysis. As a result, the final upper bound applies equally to $\sup_{\pi\in\Pi}\Vert\hat{\mathbf{T}}^{\pi}(v)-\mathcal{T}^{\pi}(v)\Vert_{\infty}$.
>
> Next we introduce two Lemmas that lead us to conclude Theorem 3.
> > **Lemma 11:**
> Let $V_{\mathcal{P}}^{\pi}$ be the fixed point of $v=\mathcal{T}^{\pi}(v)$ and $V_{\hat{\mathcal{P}}}^{\pi}$ is the fixed point of $v=\hat{\mathbf{T}}^{\pi}(v)$, then we have
> >$$
> \sup_{\pi\in\Pi}\left\Vert V_{\hat{\mathcal{P}}}^{\pi}-V_{\mathcal{P}}^{\pi}\right\Vert_{\infty}\leq \frac{1}{1-\gamma}\sup_{\pi\in\Pi}\left\Vert\hat{\mathbf{T}}^{\pi}(V_{\hat{\mathcal{P}}}^{\pi})-\mathcal{T}^{\pi}(V_{\mathcal{P}}^{\pi})\right\Vert_{\infty}
> $$
>
> > **Lemma 12:**
> >Let $\hat{\pi}^\*:=\arg\max_{\pi}V_{\hat{\mathcal{P}}}^{\pi}$, we have
> >$$0\leq \sup_{\pi\in\Pi}V_{\mathcal{P}}^{\pi}-V_{\mathcal{P}}^{\hat{\pi}^\*}\leq 2\sup_{\pi\in\Pi}\left\Vert V_{\hat{\mathcal{P}}}^{\pi}-V_{\mathcal{P}}^{\pi}\right\Vert_{\infty}$$
>
> With Lemma 11 and 12, we have that under KL-divergence uncertainty set, with probability $1-\eta$
>
> $$
> \begin{align}
> \sup_{\pi\in\Pi}V_{\mathcal{P}}^{\pi}-V_{\mathcal{P}}^{\hat{\pi}^\*}&\leq2\sup_{\pi\in\Pi}\left\Vert V_{\hat{\mathcal{P}}}^{\pi}-V_{\mathcal{P}}^{\pi}\right\Vert_{\infty}\\\\
> &\leq\frac{2}{1-\gamma}\sup_{\pi\in\Pi}\left\Vert\hat{\mathbf{T}}^{\pi}(V_{\hat{\mathcal{P}}}^{\pi})-\mathcal{T}^{\pi}(V_{\mathcal{P}}^{\pi})\right\Vert_{\infty}\\\\
> &\leq\frac{18\Vert R+\gamma v\Vert_\infty}{\sqrt{n\mathfrak{p}}(1-\gamma)}\sqrt{\log{(4|\mathcal{S}|^2|\mathcal{A}|/\eta)}}\\\\
> &\leq\frac{18}{\sqrt{n\mathfrak{p}}(1-\gamma)^2}\sqrt{\log{(4|\mathcal{S}|^2|\mathcal{A}|/\eta)}}
> \end{align}
> $$
>
> under $f_k$-divergence uncertainty set, with probability $1-\eta$
>
> $$
> \begin{align}
> \sup_{\pi\in\Pi}V_{\mathcal{P}}^{\pi}-V_{\mathcal{P}}^{\hat{\pi}^\*}&\leq2\sup_{\pi\in\Pi}\left\Vert V_{\hat{\mathcal{P}}}^{\pi}-V_{\mathcal{P}}^{\pi}\right\Vert_{\infty}\\\\
> &\leq\frac{6\cdot2^{k^\*}k^\*}{\sqrt{n\mathfrak{p}}(1-\gamma)^2}\sqrt{\log\left(4|S|^2|\mathcal{A}|/\eta\right)}
> \end{align}
> $$
> This completes the proof of Theorem 3.
>
> We then prove the claimed lemmas as follows.
>
> **Proof of Lemma 11**
>
> Similarly to Proposition 1, we first prove that $\mathcal{T}^{\pi}$ and $\hat{\mathbf{T}}^{\pi}$ are $\gamma$-contraction operators.
> $$
> \begin{align}
> \left|\mathcal{T}^{\pi}(v_1)(s)-\mathcal{T}^{\pi}(v_2)(s)\right|&=\left|\inf_{P\in\mathcal{P}}f(v_1)(s)-\inf_{P\in\mathcal{P}}f(v_2)(s)\right|\\\\
> &\leq\sup_{P\in\mathcal{P}}\left|f(v_1)(s)-f(v_2)(s)\right|\\\\
> &=\sup_{P\in\mathcal{P}}\gamma\left|\sum_{a,s'}\pi(a|s)P_{s,a}(s')v_1(s')-\sum_{a,s'}\pi(a|s)P_{s,a}(s')v_2(s')\right|\\\\
> &\leq\gamma\Vert P_s\Vert_{\infty}\Vert v_1-v_2\Vert_{\infty}\\\\
> &=\gamma\Vert v_1-v_2\Vert_{\infty}
> \end{align}
> $$
> where $\Vert P_s\Vert_\infty=\sup_{\Vert v\Vert_\infty=1}\Vert P_sv\Vert_\infty$ is the induced operator norm. By replacing $P_s$ with $P_{s,n}$, we obtain
> $$
> \left|\hat{\mathbf{T}}^{\pi}(v_1)(s)-\hat{\mathbf{T}}^{\pi}(v_2)(s)\right|\leq\gamma\Vert v_1-v_2\Vert_{\infty}
> $$
> Following the proof of Proposition 1, which only requires that $V_{\mathcal{P}}^{\pi}$ and $V_{\hat{\mathcal{P}}}^{\pi}$ are fixed points of $v = \mathcal{T}^{\pi}(v)$ and $v = \hat{\mathbf{T}}^{\pi}(v)$, respectively, and that $\mathcal{T}^{\pi}$ and $\hat{\mathbf{T}}^{\pi}$ are $\gamma$-contraction operators, we then obtain
> $$
> \sup_{\pi\in\Pi}\left\Vert V_{\hat{\mathcal{P}}}^{\pi}-V_{\mathcal{P}}^{\pi}\right\Vert_{\infty}\leq \frac{1}{1-\gamma}\sup_{\pi\in\Pi}\left\Vert\hat{\mathbf{T}}^{\pi}(V_{\hat{\mathcal{P}}}^{\pi})-\mathcal{T}^{\pi}(V_{\mathcal{P}}^{\pi})\right\Vert_{\infty}
> $$
>
> **Proof of Lemma 12**
>
> Note that
> $$
> \begin{align}
> \sup_{\pi\in\Pi}V_{\mathcal{P}}^{\pi}-V_{\mathcal{P}}^{\hat{\pi}^\*}&=\sup_{\pi\in\Pi}V_{\mathcal{P}}^{\pi}-\sup_{\pi\in\Pi}V_{\hat{\mathcal{P}}}^\pi+V_{\hat{\mathcal{P}}}^{\hat{\pi}^\*}-V_{\mathcal{P}}^{\hat{\pi}^\*}\\\\
> &\leq\left\Vert\sup_{\pi\in\Pi}V_{\mathcal{P}}^{\pi}-\sup_{\pi\in\Pi}V_{\hat{\mathcal{P}}}^\pi\right\Vert_{\infty}+\left\Vert V_{\hat{\mathcal{P}}}^{\hat{\pi}^*}-V_{\mathcal{P}}^{\hat{\pi}^\*}\right\Vert_{\infty}\\\\
> &\leq2\sup_{\pi\in\Pi}\left\Vert V_{\mathcal{P}}^{\pi}-V_{\hat{\mathcal{P}}}^\pi\right\Vert_{\infty}
> \end{align}
> $$
> ### Q2. The involvement of the minimum support probability for $f_k$-divergence uncertainty set is sub-optimal when compared to [Theorem 3.2(b), 1].
>
> In our analysis, we  focus on a $\rho$-free bound that holds uniformly for all $k$. We believe that the minimum support probability should be present in the bound as $k \to 1$. This is reflected in [3], as without such a minimum support probability assumption, it is not possible to achieve the $n^{-1/2}$ rate in a minimax sense. In contrast, [Theorem 3.2(b), 1] considers only the $\chi^2$-divergence case, and yields a looser bound in terms of its dependence on $\rho$ as $\rho \to 0$. Moreover, their dependence on $\mathfrak{p}$ is worse than ours in the KL-divergence setting. We believe that a $\mathfrak{p}$-free bound for the $\chi^2$-divergence may be attainable through another specialized analysis.
>
> ### Q3. Can you explain why the bounds in Theorems 1 and 2 do not involve the uncertainty radius $\rho$?
>
> We thank the reviewer for the insightful comments. Intuitively, as $\rho = 0$ represents the non-robust MDP setting, the sample complexity of robust policy learning should not be unbounded as $\rho\rightarrow 0$. This behaviour of the sample complexity’s independence of $\rho$ as $\rho\rightarrow 0$ is first observed in [4] in the SA rectangular setting with KL divergence, and we generalize to the more challenging S-rectangular setting with both KL and $f_k$ divergences, where the SOTA bounds all have a $1/\rho$ dependence [1].
>
> Technically, for the KL‑divergence, the uncertainty radius $\rho$ is canceled out in the following calculation since $f$ contains $-\lambda|\mathcal{A}|\rho$:
>
> $$
> |f(\boldsymbol{\mu}_n,u,\lambda)-f(\boldsymbol{\mu},u,\lambda)|
> $$
>
> For the $f_k$‑divergence, according to Lemma 9 (D.3), $c(k,\rho,|\mathcal{A}|)$ can be expressed in terms of $\mu$ and $w$:
>
> $$
> \left(\sum_{a\in\mathcal{A}}\mu_a\left[w_a^{k^*}\right]\right)^{1/k}
> = c(k,\rho,|\mathcal{A}|)\mu_i\left[w_i^{1/(k-1)}\right],
> \quad \text{for all } i \in \mathcal{A}. \tag{D.3}
> $$
> Therefore, $\rho$ no longer appears in the subsequent analysis or the final bounds.
>
> [1] Yang, W., Zhang, L., & Zhang, Z. (2022). Toward theoretical understandings of robust markov decision processes: Sample complexity and asymptotics. The Annals of Statistics, 50(6), 3223-3248.
>
> [2] Clavier, P., Shi, L., Le Pennec, E., Mazumdar, E., Wierman, A., & Geist, M. (2024). Near-Optimal Distributionally Robust Reinforcement Learning with General $ L_p $ Norms. Advances in Neural Information Processing Systems, 37, 1750-1810.
>
> [3] Duchi, J. C., & Namkoong, H. (2021). Learning models with uniform performance via distributionally robust optimization. The Annals of Statistics, 49(3), 1378-1406.
>
> [4] Wang, S., Si, N., Blanchet, J., & Zhou, Z. (2024). Sample complexity of variance-reduced distributionally robust Q-learning. Journal of Machine Learning Research, 25(341), 1-77.

---

> > ### Comment · Reviewer_5h1f · 2025-08-04
> >
> > Thank you for your response. I can understand how Theorems 1 and 2 can translate to sample complexity bounds. On the other hand, I still find it a bit counterintuitive that these bounds do not depend on the uncertainty radius. This seems to suggest that the proposed algorithm requires the same amount of data regardless of how uncertain it is about the reference model. However, when $\rho$ becomes (extremely) large, the sample complexity is expected to be lower.

---

> > > ### Author Response · Authors · 2025-08-04
> > > **Thanks for the questions**
> > >
> > > Thank you for your question. When $\rho$ is large, our sample complexity upper bound is indeed not optimal. However, the regime with large $\rho$ is not practically interesting, since in this case the model can hardly utilize any prior information, making the setting overly conservative and rarely encountered in real-world applications.

---

> ### Author Response · Authors · 2025-08-05
>
> Dear Reviewer 5h1f,
>
> We hope that we have addressed all of your concerns. As the author-reviewer discussion period concludes, please do not hesitate to reach out if further clarifications are needed. We would be delighted to provide additional explanations or adjustments.
>
> Thank you again for your invaluable feedback and contributions to this work.
>
> Best wishes,
> The authors

---

### Official Review · Reviewer_G8HC · 2025-07-03

**Clarity:** 3
**Significance:** 2
**Originality:** 3
**Rating:** 3
**Confidence:** 3

**Summary:**

This paper focuses on divergence-based S-rectangular distributionally robust reinforcement learning (DR-RL), with the core being establishing near-optimal sample complexity bounds under this framework and validating their effectiveness through theoretical analysis and numerical experiments.

**Questions:**

Suggestions:
1.The authors should supplement additional numerical experiments to systematically investigate the impact of key hyperparameters on the estimation error and sample complexity
2.The authors should elaborate on how the established near-optimal sample complexity theory directly guides the practical implementation of S-rectangular divergence-based DR-RL algorithms.

**Ethical Concerns:**

["NO or VERY MINOR ethics concerns only"]

**Final Justification:**

I thank the authors for addressing my concerns. The work fills a key gap in the sample complexity analysis of S-rectangular models, with clear contributions, though its practical applicability is limited. I have updated my score accordingly.

**Limitations:**

Yes

**Quality:**

2

**Strengths And Weaknesses:**

Strengths
-The article considers a reward function $$R(S_t, A_t, S_{t+1})$$ that depends on the "current state, current action, and next state", a structure often overlooked in existing research.
- The article conducts numerical experiments on a robust inventory control problem and a theoretical worst-case example, verifying that the sample complexity decays at a rate of $$1/\sqrt{n}$$ with respect to the sample size n and has a linear dependence on $$|S||A|$$, which is consistent with theoretical deductions, enhancing the credibility of the conclusion.
Weaknesses：
1.The numerical experiments only run under a single set of hyperparameters without reporting error bars or statistical significance information, making it difficult to fully verify the robustness of the results.
2.The research primarily focuses on the theoretical analysis of sample complexity, with little discussion of practical implementation challenges. This narrow focus limits its guidance for real-world applications.

---

> ### Author Rebuttal · Authors · 2025-07-31
>
> We thank the reviewer for their comments and valuable suggestions. We address their questions below.
>
> **W&Q1.Numerical experiments**
>
> We thank the reviewer for the valuable suggestion for additional numerical validation. In response to the reviewer’s concerns about the comprehensiveness and robustness of the experiments, we would like to clarify two key points:
>
> First, our algorithm does not contain hyper‑parameters that require tuning; the inputs to the algorithm are all known quantities that do not require prior knowledge of the MDP instance.
>
> Second, regarding the original submission, we performed only one run on a benchmark instance. There was no repetition and hence error bars could not be produced. Our figure demonstrates remarkably low variability in the performance of our algorithm: the log-log plot reveals an error convergence rate with a slope of −0.5 even without averaging. However, your suggestion to include an error bar diagram could improve the clarity. We implemented this in our revision and produced the tables below.
>
> In the revised manuscript, to further verify the performance of our method, we have introduced additional empirical validation on multiple instances with different environment parameters (e.g., price, purchase cost, etc.). For each instance, we conduct **200 independent runs with different random seeds**, where the only source of randomness is the sample set drawn from the demand distribution. The aggregated results for these instances are reported below.
>
>
> **instance 1** (original instance in the paper)
> $I=10,B=5,O=5,p=3,c=2,h=0.2,b=3,\gamma =0.9, P_D=[0.1,0.2,0.3,0.3,0.1]$ with KL divergence unceratinty sets.
>
> |     n     |   mean   | 2.5th percentile | 97.5th percentile |
> | :-------: | :------: | :--------------: | :---------------: |
> |    960    | 6.125496 |     2.831040     |     10.756915     |
> |   9600    | 1.068351 |     0.452896     |     2.103853      |
> |   96000   | 0.314591 |     0.124520     |     0.584948      |
> |  960000   | 0.099442 |     0.042141     |     0.206225      |
> |  9600000  | 0.030685 |     0.013754     |     0.055064      |
> | 96000000  | 0.010317 |     0.004061     |     0.020621      |
> | 960000000 | 0.003025 |     0.001341     |     0.006231      |
>
> **Instance 2**: New instance (low back order cost setting)
> $I=10,B=5,O=5,p=3,c=2,h=0.2,b=1,\gamma =0.9, P_D=[0.1,0.2,0.3,0.3,0.1]$  with KL divergence unceratinty sets.
>
>
> |    n     |   mean   | 2.5th percentile | 97.5th percentile |
> | :------: | :------: | :--------------: | :---------------: |
> |   960    | 6.185270 |     3.055882     |     10.811357     |
> |   9600   | 1.103974 |     0.475384     |     2.249074      |
> |  96000   | 0.327697 |     0.132400     |     0.631491      |
> |  960000  | 0.106403 |     0.040831     |     0.209798      |
> | 9600000  | 0.033056 |     0.013672     |     0.059421      |
> | 96000000 | 0.010739 |     0.004110     |     0.021523      |
>
>
> **Instance 3**: New instance (high holding cost setting)
>
> $I=10,B=5,O=5,p=3,c=2,h=1,b=3, \gamma =0.9, P_D=[0.1,0.2,0.3,0.3,0.1]$ with KL divergence unceratinty sets.
>
>
> |    n     |   mean    | 2.5th percentile | 97.5th percentile |
> | :------: | :-------: | :--------------: | :---------------: |
> |   960    | 10.388948 |     4.918980     |     19.464402     |
> |   9600   | 2.465194  |     1.034968     |     5.128030      |
> |  96000   | 0.736195  |     0.298026     |     1.405253      |
> |  960000  | 0.235467  |     0.092150     |     0.499871      |
> | 9600000  | 0.074945  |     0.033755     |     0.134991      |
> | 96000000 | 0.023835  |     0.010401     |     0.050705      |
>
>
>
>
>
> **W&Q2. Guidance to the practice**
>
>
> We would also like to clarify that both the statistical analysis and optimization procedure are important factors in achieving a practical algorithm. While there are many existing studies on how to efficiently compute the optimal policy given an S-rectangular robust MDP model under $f_k$ and KL divergence from an optimization point of view [1,2], the statistical complexity upper bounds in the literature have been overly pessimistic, presenting a suboptimal $|\mathcal{S}|^2$ dependence. In contrast, our work marks the first contribution that exhibits a linear sample complexity dependence on the number of states $|\mathcal{S}|$ in the $f_k$ and KL divergence settings. This improved dependence elicits the fact that  achieving robust policy learning is not statistically more challenging than in the classical MDP setting. Therefore, coupled with the fact that $f_k$ and KL divergence are the most commonly used models in practice, our sample complexity results can guide the design of practical robust algorithms that are as sample-efficient as non-robust algorithms.
>
> [1] Ho, C. P., Petrik, M., & Wiesemann, W. (2018, July). Fast Bellman updates for robust MDPs. In *International Conference on Machine Learning* (pp. 1979-1988). PMLR.
>
> [2] Ho, C. P., Petrik, M., & Wiesemann, W. (2022). Robust $\phi $-Divergence MDPs. *Advances in Neural Information Processing Systems*, *35*, 32680-32693.

---

> ### Author Response · Authors · 2025-08-05
> **Further comments**
>
> Dear Reviewer G8HC,
>
> We sincerely appreciate the time and effort you have dedicated to reviewing our manuscript and providing valuable feedback.
>
> As the discussion phase ends soon (by 8.6), we hope our responses have addressed your concerns. If you need further clarification or have any additional concerns, we are willing to continue the communication with you.
>
> Best regards,
> The Authors

---

### Note · Authors · 2025-08-12

We thank the reviewers for their thoughtful feedback. We would like to clarify the distinction between statistical complexity and iteration complexity, as well as the core technical challenges addressed in our paper.

**Statistical complexity vs. iteration complexity**
Statistical complexity concerns how many training samples $n$ are required for the learned model to be close to the optimal model under true distribution, whereas iteration complexity concerns, given a fixed dataset, how many iterations are needed for the algorithm to converge to the empirical optimum. The two concepts are fundamentally different and not directly comparable.

**Technical challenges**
Prior work on divergence-based S-rec robust MDP [1] suffers from: (i) an undesirable $1/\rho$ dependence on the robustness radius, and (ii) quadratic or worse dependence on $|\mathcal S|$ and $|\mathcal A|$. For  $L_p$-norm uncertainty, [2] achieved linear scaling and avoided $1/\rho$, but their techniques do not extend to divergence-based sets.

Compared with the non-robust setting, the robust transition kernel $P$ ranges over an uncertainty set $\mathcal P$, and the optimal policy may be randomized. This invalidates the usual Q-function–based route (which typically targets deterministic policies). Consequently, one must analyze a richer policy class that includes all randomized policies, substantially enlarging the effective hypothesis space.

[1] Yang, Wenhao, Liangyu Zhang, and Zhihua Zhang. "Toward theoretical understandings of robust Markov decision processes: Sample complexity and asymptotics." *Annals of Statistics*, 50(6), 2022.

[2] Clavier, P., Shi, L., Le Pennec, E., Mazumdar, E., Wierman, A., & Geist, M. (2024). Near-Optimal Distributionally Robust Reinforcement Learning with General Norms. Advances in Neural Information Processing Systems.

---

### Decision · Program_Chairs · 2025-09-17

**Decision:**

Reject

**Comment:**

This paper develops sample complexity bounds for divergence-based S-rectangular DR-RL that achieve linear scaling with the size of the state and action spaces. Prior literature has developed divergence-based SA-rectangular sample complexity and Lp-based S-rectangular sample complexity, and this paper fills the gap in the said setting. While the theoretical results are valuable, it falls short in several aspects from being accepted: 1) justify the near-optimality by providing rigorous lower bounds, over the entire range of the uncertainty level. The said bound in the abstract ignores the key parameter (minimum support probability), which is misleading. It is also unclear if the dependency with the discount factor is likely sub-optimal. 2) Better position the sample complexity with existing works and highlight the novelty in terms of analysis - in particular, why the Lp based/chi-square result cannot be extended straightforwardly.

The authors have also added new experiments and additional results, which should be incorporated in the future versions.